# Infinite Recommendation Networks: A Data-Centric Approach

**Noveen Sachdeva**[†]     **Mehak Preet Dhaliwal**[†]     **Carole-Jean Wu**[‡]     **Julian McAuley**[†]

University of California, San Diego[†]    Meta AI[‡]

{nosachde,mdhaliwal,jmcauley}@ucsd.edu
carolejeanwu@meta.com

## Abstract

We leverage the Neural Tangent Kernel and its equivalence to training infinitely-wide neural networks to devise $\infty$-AE: an autoencoder with infinitely-wide bottle-neck layers. The outcome is a highly expressive yet simplistic recommendation model with a single hyper-parameter and a closed-form solution. Leveraging $\infty$-AE's simplicity, we also develop DISTILL-CF for synthesizing tiny, high-fidelity data summaries which distill the most important knowledge from the extremely large and sparse user-item interaction matrix for efficient and accurate subsequent data-usage like model training, inference, architecture search, *etc.* This takes a data-centric approach to recommendation, where we aim to improve the quality of logged user-feedback data for subsequent modeling, independent of the learning algorithm. We particularly utilize the concept of differentiable Gumbel-sampling to handle the inherent data heterogeneity, sparsity, and semi-structuredness, while being scalable to datasets with hundreds of millions of user-item interactions. Both of our proposed approaches significantly outperform their respective state-of-the-art and when used together, we observe $96 - 105\%$ of $\infty$-AE's performance on the full dataset with as little as $0.1\%$ of the original dataset size, leading us to explore the counter-intuitive question: *Is more data what you need for better recommendation?*

## 1   Introduction

The Neural Tangent Kernel (NTK) has recently advanced the theoretical understanding of how neural networks learn [2, 20]. Notably, performing Kernelized Ridge Regression (KRR) with the NTK has been shown to be equivalent to training infinitely-wide neural networks for an infinite number of SGD steps. Owing to KRR's closed-form solution, these networks can be trained in a fast and efficient manner whilst not sacrificing expressivity. While the application of infinite neural networks is being explored in various learning problems [48, 3], detailed comparative analyses demonstrate that deep, finite-width networks tend to perform significantly better than their infinite-width counterparts for a variety of standard computer-vision tasks [31].

On the contrary, for recommendation tasks, there always has been a debate of linear *vs.* non-linear networks [29, 65], along with the importance of increasing the width *vs.* depth of the network [11, 39]. At a high level, the general conclusion is that a well-tuned, wide and linear network can outperform shallow and deep neural networks for recommendation [50]. We extend this debate by introducing our model $\infty$-AE, an autoencoder with infinitely wide bottleneck layers and examine its behavior on the recommendation task. Our evaluation demonstrates significantly improved results over state-of-the-art (SoTA) models across various datasets and evaluation metrics.

36th Conference on Neural Information Processing Systems (NeurIPS 2022).

A rising challenge in recommender systems research has been the increased cost of training models on massive datasets which can involve billions of user-item interaction logs, in terms of time, computational resources, as well as the downstream carbon footprint. Moreover, despite anonymization efforts, privacy risks have been associated with publicly released user data [38]. To this end, we further explore recommendation from a data-centric viewpoint [60], which we loosely define as:

**Definition 1.1. (Data-centric AI)** *Systematic methods for improving the collected data's quality, thereby shifting the focus from merely acquiring large quantities of data; implicitly helping in a learning algorithm's generalization, scalability, and eco-sustainability.*

Previous work on data-centric techniques generally involve constructing terse data summaries of large datasets, and has focused on domains with continuous, real-valued features such as images [66, 41], which are arguably more amenable to data optimization approaches. Due to the heterogeneity, sparsity, and semi-structuredness of recommendation data, such methods are not directly applicable. Common approaches for scaling down such recommendation datasets typically include heuristics such as random, head-user, or k-core sampling. More principled approaches include coreset construction [57] that focus on optimizing for *picking* the set of data-points that are the most representative from a given dataset, and are generally shown to out-perform various heuristic sampling strategies. However, *synthesizing* informative summaries for recommendation datasets still remains a challenge.

Consequently, we propose DISTILL-CF, a data distillation framework for collaborative filtering (CF) datasets that utilizes $\infty$-AE in a bilevel optimization objective to create highly compressed, informative, and generic data summaries. DISTILL-CF employs efficient multi-step differentiable Gumbel-sampling [23] at each step of the optimization to encompass the heterogeneity, sparsity, and semi-structuredness of recommendation data. We further provide an analysis of the denoising effect observed when training the model on the synthesized versus the full dataset.

To summarize, *in this paper*, we make the following contributions:

- We develop $\infty$-AE: an infinite-width autoencoder for recommendation, that aims to reconstruct the incomplete preferences in a user's item consumption sequence. We demonstrate its efficacy on four datasets, and demonstrate that $\infty$-AE outperforms complicated SoTA models with only a single fully-connected layer, closed-form optimization, and a single hyper-parameter. We believe our work to be the first to demonstrate that an infinite-width network can outperform their finite-width SoTA counterparts for practical scenarios like recommendation.

- We subsequently develop DISTILL-CF: a novel data distillation framework that can synthesize tiny yet accurate data summaries for a variety of modeling applications. We empirically demonstrate similar performance of models trained on the full dataset versus training the same models on $2 - 3$ orders smaller data summaries synthesized by DISTILL-CF. Notably, DISTILL-CF and $\infty$-AE are complementary for each other's practicality, as $\infty$-AE's closed-form solution enables DISTILL-CF to scale to datasets with hundreds of millions of interactions; whereas, DISTILL-CF's succinct data summaries help improving $\infty$-AE's restrictive training complexity, and achieving SoTA performance when trained on these tiny data summaries.

- Finally, we also note that DISTILL-CF has a strong data denoising effect, validated with a counter-intuitive observation — if there is noise in the original data, models trained on *less* data synthesized by DISTILL-CF can be better than the same model trained on the *entire* original dataset. Such observations, along with the strong data compression results attained by DISTILL-CF, reinforce our data-centric viewpoint to recommendation, encouraging the community to think more about the quality of collected data, rather than its quantity.

## 2 Related Work

**Autoencoders for recommendation.** Recent approaches to implicit feedback recommendation involve building models that can re-construct an incomplete user preference list using autoencoders [32, 59, 54, 33]. The CDAE model [64] first introduced this idea and used a standard denoising autoencoder for recommending new items to users. MVAE [32] later extended this idea to use variational autoencoders, and provided a principled approach to perform variational inference for this model architecture. More recently, EASE [59] proposed using a shallow autoencoder and estimates only an item-item similarity matrix by performing ordinary least squares regression on the relevance matrix, resulting in closed-form optimization.

**Infinite neural networks.** The Neural Tangent Kernel (NTK) [20] has gained significant attention because of its equivalence to training infinitely-wide neural networks by performing KRR with the NTK. Recent work also demonstrated the double-descent risk curve [4] that extends the classical regime of train *vs.* test error for overparameterized neural networks, and plays a crucial role in the generalization capabilities of such infinite neural networks. However, despite this emerging promise of utilizing NTK for learning problems, detailed comparative analyses [43, 31, 2] for computer vision tasks demonstrate that finite-width networks are still significantly more accurate than infinite-width ones. Looking at recommendation systems, [65] performed a theoretical comparison between Matrix Factorization (MF) and Neural MF [18] by studying their expressivity in the infinite-width limit, comparing the NTK of both of these algorithms. Notably, their settings involved the typical (user-ID, item-ID) input to the recommendation model, and observed results that were equivalent to a random predictor. [48] performed a similar study that performed matrix completion using the NTK of a single layer fully-connected neural network, but assumed meaningful feature-priors available beforehand.

**Data sampling & distillation.** Data sampling is ubiquitous — sampling negatives while contrastive learning [51, 25], sampling large datasets for fast experimentation [57], sampling for evaluating expensive metrics [30], *etc.* In this paper, we primarily focus on sampling at the dataset level, principled approaches of which can be categorized into: (1) coreset construction methods that aim to *pick* the most useful datapoints for subsequent model training [7, 27, 53, 26]. These methods typically assume the availability of a submodular set-function $f : \mathbf{V} \mapsto \mathbb{R}_+ \ \forall \ \mathbf{V} \subseteq \mathbf{X}$ for a given dataset $\mathbf{X}$ (see [6] for a systematic review on submodularity), and use this set-function $f$ as a proxy to guide the search for the most informative subset; and (2) dataset distillation: instead of picking the most informative data-points, dataset distillation techniques aim to *synthesize* data-points that can accurately distill the knowledge from the entire dataset into a small, synthetic data summary. Originally proposed in [62], the authors built upon the notions of gradient-based hyper-parameter optimization [34] to synthesize representative images for model training. Subsequently, a series of works [67, 66, 40, 41] propose various subtle modifications to the original framework, for improving the sample-complexities of models trained on data synthesized using their algorithms. Note that such distillation techniques focused on continuous data like images, which are trivial to optimize in the original data-distillation framework. More recently, [24] proposed a distillation technique for synthesizing fake graphs, but also assumed to have innate node representations available beforehand, prohibiting their method's application for recommendation data.

## 3   ∞-AE: Infinite-width Autoencoders for Recommendation

**Notation.** Given a recommendation dataset $\mathcal{D} := \{(\text{user}_i, \text{item}_i, \text{relevance}_i)\}_{i=1}^n$ consisting of $n$ user-item interactions defined over a set of users $\mathcal{U}$, set of items $\mathcal{I}$, and operating over a binary relevance score (implicit feedback); we aim to best model user preferences for item recommendation. The given dataset $\mathcal{D}$ can also be viewed as an interaction matrix, $X \in \mathbb{R}^{|\mathcal{U}| \times |\mathcal{I}|}$ where each entry $X_{u,i}$ either represents the observed relevance for item $i$ by user $u$ or is missing. Note that $X$ is typically extremely sparse, *i.e.*, $n \ll |\mathcal{U}| \times |\mathcal{I}|$. More formally, we define the problem of recommendation as accurate likelihood modeling of $\text{P}(i \mid u, \mathcal{D}) \ \forall u \in \mathcal{U}, \ \forall i \in \mathcal{I}$.

**Model.** ∞-AE takes an autoencoder approach to recommendation, where the all of the bottleneck layers are infinitely-wide. Firstly, to make the original bi-variate problem of which *item* to recommend to which *user* more amenable for autoencoders, we make a simplifying assumption that a user can be represented only by their historic interactions with items, *i.e.*, the much larger set of users lie in the linear span of items. This gives us two kinds of modeling advantages: (1) we no longer have to find a unique latent representation of users; and (2) allows ∞-AE to be trivially applicable for any user not in $\mathcal{U}$. More specifically, for a given user $u$, we represent it by the sparse, bag-of-words vector of historical interactions with items $X_u \in \mathbb{R}^{|\mathcal{I}|}$, which is simply the $u^{\text{th}}$ row in $X$. We then employ the Neural Tangent Kernel (NTK) [20] of an autoencoder that takes the bag-of-items representation of users as input and aims to reconstruct it. Due to the infinite-width correspondence [20, 2], performing Kernelized Ridge Regression (KRR) with the estimated NTK is equivalent to training an infinitely-wide autoencoder for an infinite number of SGD steps. More formally, given the NTK, $\mathbb{K} : \mathbb{R}^{|\mathcal{I}|} \times \mathbb{R}^{|\mathcal{I}|} \mapsto \mathbb{R}$ over an RKHS $\mathcal{H}$ of a single-layer autoencoder with an activation function $\sigma$ (see [47] for the NTK derivation of a fully-connected neural network), we reduce the recommendation problem to KRR as follows:

$$\underset{[\alpha_j]_{j=1}^{|\mathcal{U}|}}{\arg\min} \quad \sum_{u \in \mathcal{U}} \|f(X_u \mid \alpha) - X_u\|_2^2 + \lambda \cdot \|f\|_{\mathcal{H}}^2$$

$$\text{s.t.} \quad f(X_u \mid \alpha) = \sum_{j=1}^{|\mathcal{U}|} \alpha_j \cdot \mathbb{K}(X_u, X_{u_j}) \quad ; \quad \mathbb{K}(X_u, X_v) = \tilde{\sigma}(X_u^T X_v) + \tilde{\sigma}'(X_u^T X_v) \cdot X_u^T X_v \tag{1}$$

Where $\lambda$ is a fixed regularization hyper-parameter, $\alpha := [\alpha_1; \alpha_2 \cdots ; \alpha_{|\mathcal{U}|}] \in \mathbb{R}^{|\mathcal{U}| \times |\mathcal{I}|}$ are the set of dual parameters we are interested in estimating, $\tilde{\sigma}$ represents the dual activation of $\sigma$ [14], and $\tilde{\sigma}'$ represents its derivative. Defining a gramian matrix $K \in \mathbb{R}^{|\mathcal{U}| \times |\mathcal{U}|}$ where each element can intuitively be seen as the *similarity* of two users, *i.e.*, $K_{i,j} := \mathbb{K}(X_{u_i}, X_{u_j})$, the optimization problem defined in Equation (1) has a closed-form solution given by $\hat{\alpha} = (K + \lambda I)^{-1} \cdot X$. We can subsequently perform inference for any novel user as follows: $\hat{P}(\cdot \mid u, \mathcal{D}) = \text{softmax}(f(X_u \mid \hat{\alpha}))$. We also provide $\infty$-AE's training and inference pseudo-codes in Appendix A, Algorithms 1 and 2.

**Scalability.** We carefully examine the computational cost of $\infty$-AE's training and inference. Starting with training, $\infty$-AE has the following computationally-expensive steps: (1) computing the gramian matrix $K$; and (2) performing its inversion. The overall training time complexity thus comes out to be $\mathcal{O}(|\mathcal{U}|^2 \cdot |\mathcal{I}| + |\mathcal{U}|^{2.376})$ if we use the Coppersmith-Winograd algorithm [12] for matrix inversion, whereas the memory complexity is $\mathcal{O}(|\mathcal{U}| \cdot |\mathcal{I}| + |\mathcal{U}|^2)$ for storing the data matrix $X$ and the gramian matrix $K$. As for inference for a single user, both the time and memory requirements are $\mathcal{O}(|\mathcal{U}| \cdot |\mathcal{I}|)$. Observing these computational complexities, we note that $\infty$-AE can be difficult to scale-up to larger datasets naïvely. To this effect, we address $\infty$-AE's scalability challenges in DISTILL-CF (Section 4), by leveraging a simple observation from support vector machines: not all datapoints (users in our case) are important for model learning. Additionally, in practice, we can perform all of these matrix operations on accelerators like GPU/TPUs and achieve a much higher throughput.

## 4  DISTILL-CF

**Motivation.** Representative sampling of large datasets has numerous downstream applications. Consequently, in this section we develop DISTILL-CF: a data distillation framework to *synthesize* small, high-fidelity data summaries for collaborative filtering (CF) data. We design DISTILL-CF with the following rationales: (1) mitigating the scalability challenges in $\infty$-AE by training it only on the terse data summaries generated by DISTILL-CF; (2) improving the sample complexity of existing, finite-width recommendation models; (3) addressing the privacy risks of releasing user feedback data by releasing their synthetic data summaries instead; and (4) abating the downstream $CO_2$ emissions of frequent, large-scale recommendation model training by estimating their parameters only on much smaller data summaries synthesized by DISTILL-CF.

**Challenges.** Existing work in data distillation has focused on continuous domain data such as images [40, 41, 67, 66], and are not directly applicable to heterogeneous and semi-structured domains such as recommender systems and graphs. This problem is further exacerbated since the data for these tasks is severely sparse. For example, in recommendation scenarios, a vast majority of users interact with very few items [22]. Likewise, the nodes in a number of graph-based datasets tend to have connections with very small set of nodes [68]. We will later show how our DISTILL-CF framework is elegantly able to circumvent both these issues while being accurate, and scalable to large datasets.

**Methodology.** Given a recommendation dataset $\mathcal{D}$, we aim to synthesize a smaller, support dataset $\mathcal{D}^s$ that can match the performance of recommendation models $\phi : \mathcal{U} \times \mathcal{I} \mapsto \mathbb{R}$ when trained on $\mathcal{D}$ versus $\mathcal{D}^s$. We take inspiration from [40], which is also a data distillation technique albeit for images. Formally, given a recommendation model $\phi$, a held-out validation set $\mathcal{D}^{\text{val}}$, and a differentiable loss function $l : \mathbb{R} \times \mathbb{R} \mapsto \mathbb{R}$ that measures the correctness of a prediction with the actual user-item relevance, the data distillation task can be viewed as the following bilevel optimization problem:

$$\underset{\mathcal{D}^s}{\arg\min} \quad \underset{(u,i,r) \sim \mathcal{D}^{\text{val}}}{\mathbb{E}} [l(\phi_{\mathcal{D}^s}^*(u,i), r)] \quad ; \quad \text{s.t.} \quad \phi_{\mathcal{D}^s}^* := \underset{\phi}{\arg\min} \quad \underset{(u,i,r) \sim \mathcal{D}^s}{\mathbb{E}} [l(\phi(u,i), r)] \tag{2}$$

This optimization problem has an *outer loop* which searches for the most informative support dataset $\mathcal{D}^s$ given a fixed recommendation model $\phi^*$, whereas the *inner loop* aims to find the optimal recommendation model for a fixed support set. In DISTILL-CF, we use $\infty$-AE as the model of choice at each step of the inner loop for two reasons: (1) as outlined in Section 3, $\infty$-AE has a closed-form solution with a single hyper-parameter $\lambda$, making the inner-loop extremely efficient; and (2) due to the infinite-width correspondence [20, 2], $\infty$-AE is equivalent to training an infinitely-wide autoencoder on $\mathcal{D}^s$, thereby not compromising on performance.

For the outer loop, we focus only on synthesizing *fake users* (given a fixed user budget $\mu$) through a learnable matrix $X^s \in \mathbb{R}^{\mu \times |\mathcal{I}|}$ which stores the interactions for each fake user in the support dataset. However, to handle the discrete nature of the recommendation problem, instead of directly optimizing for $X^s$, DISTILL-CF instead learns a *continuous prior* for each user-item pair, denoting the importance of sampling that interaction in our support set (similar to the notion of propensity [56, 58]). We then sample $\hat{X}^s \sim X^s$ to get our final, discrete support set.

Instead of keeping this sampling operation post-hoc, *i.e.*, after solving for the optimization problem in Equation (2); we perform differentiable Gumbel-sampling [23] on each row (user) in $X^s$ at every optimization step, thereby ensuring search only over sparse and discrete support sets. A notable property of recommendation data is that each user can interact with a variable number of items (this distribution is typically long-tailed due to Zipf's law [63]). We circumvent this dynamic user sparsity issue by taking multiple Gumbel-samples for each user, with replacement. This implicitly gives DISTILL-CF the freedom to control the user and item popularity distributions in the generated data summary $\hat{X}^s$ by adjusting the entropy in the prior-matrix $X^s$.

Having controlled for the discrete and dynamic nature of recommendation data by the multi-step Gumbel-sampling trick, we further focus on maintaining the sparsity of the synthesized data. To do so, in addition to the outer-loop reconstruction loss, we add an L1-penalty over $\hat{X}^s$ to promote and explicitly control its sparsity [16, Chapter 3]. Furthermore, tuning the number of Gumbel samples we take for each fake user, gives us more control over the sparsity in our generated data summary. More formally, the final optimization objective in DISTILL-CF can be written as:

$$\underset{X^s}{\arg\min} \quad \underset{u \sim \mathcal{U}}{\mathbb{E}} \left[ X_u \cdot \log(K_{X_u \hat{X}^s} \cdot \alpha) + (1 - X_u) \cdot \log(1 - K_{X_u \hat{X}^s} \cdot \alpha) \right] + \lambda_2 \cdot ||\hat{X}^s||_1$$

$$\text{s.t.} \quad \alpha = (K_{\hat{X}^s \hat{X}^s} + \lambda I)^{-1} \cdot \hat{X}^s \quad ; \quad \hat{X}^s = \sigma \left( \sum_{i=1}^{\gamma} \text{Gumbel}_\tau(\text{softmax}(X^s)) \right) \tag{3}$$

Where, $\lambda_2$ represents an appropriate regularization hyper-parameter for minimizing the L1-norm of the sampled support set $\hat{X}^s$, $K_{XY}$ represents the gramian matrix for the NTK of $\infty$-AE over $X$ and $Y$ as inputs, $\tau$ represents the temperature hyper-parameter for Gumbel-sampling, $\gamma$ denotes the number of samples to take for each fake user in $X^s$, and $\sigma$ represents an appropriate activation function which clips all values over 1 to be exactly 1, thereby keeping $\hat{X}^s$ binary. We use hard-tanh in our experiments. We also provide DISTILL-CF's pseudo-code in Appendix A, Algorithm 3.

**Scalability.** We now analyze the time and memory requirements for optimizing Equation (3). The inner loop's major component clearly shares the same complexity as $\infty$-AE. However, since the parameters of $\infty$-AE ($\alpha$ in Equation (1)) are now being estimated over the much smaller support set $\hat{X}^s$, the time complexity reduces to $\mathcal{O}(\mu^2 \cdot |\mathcal{I}|)$ and memory to $\mathcal{O}(\mu \cdot |\mathcal{I}|)$, where $\mu$ typically only lies in the range of hundreds for competitive performance. On the other hand, for performing multi-step Gumbel-sampling for each synthetic user, the memory complexity of a naïve implementation would be $\mathcal{O}(\gamma \cdot \mu \cdot |\mathcal{I}|)$ since AutoGrad stores all intermediary variables for backward computation. However, since the gradient of each Gumbel-sampling step is independent of other sampling steps and can be computed individually, using `jax.custom_vjp`, we reduced its memory complexity to $\mathcal{O}(\mu \cdot |\mathcal{I}|)$, adding nothing to the overall inner-loop complexity.

For the outer loop, we optimize the logistic reconstruction loss using SGD where we randomly sample a batch of $b$ users from $\mathcal{U}$ and update $X^s$ directly. In totality, for an $\xi$ number of outer-loop iterations, the time complexity to run DISTILL-CF is $\mathcal{O}(\xi \cdot (\mu^2 + b + b \cdot \mu) \cdot |\mathcal{I}|)$, and $\mathcal{O}(b \cdot \mu + (\mu + b) \cdot |\mathcal{I}|)$ for memory. In our experiments, we note convergence in only hundreds of outer-loop iterations, making DISTILL-CF scalable even for the largest of the publicly available datasets and practically useful.

Table 1: Comparison of ∞-AE with different methods on various datasets. All metrics are better when higher. Brief set of data statistics can be found in Appendix B.3, Table 2. **Bold** values represent the best in a given row, and underlined represent the second-best. Results for ∞-AE on the Netflix dataset (marked with a *) consist of random user-sampling with a max budget of $25K \equiv 5.4\%$ users, and results for DISTILL-CF + ∞-AE have a user-budget of 500 for all datasets.

| Dataset | Metric | PopRec | Bias only | MF | NeuMF | Light GCN | EASE | MVAE | ∞-AE | DISTILL-CF + ∞-AE |
|---------|--------|--------|-----------|-----|--------|-----------|------|------|------|-------------------|
| **Amazon Magazine** | AUC | 0.8436 | 0.8445 | 0.8475 | 0.8525 | 0.8141 | 0.6673 | 0.8507 | 0.8539 | **0.8584** |
| | HR@10 | 14.35 | 14.36 | 18.36 | 18.35 | 27.12 | 26.31 | 17.94 | 27.09 | **28.27** |
| | HR@100 | 59.5 | 59.35 | 58.94 | 59.3 | 58.00 | 48.36 | 57.3 | 60.86 | **61.78** |
| | NDCG@10 | 8.42 | 8.33 | 13.1 | 13.6 | 22.57 | 22.84 | 12.18 | 23.06 | **23.81** |
| | NDCG@100 | 19.38 | 19.31 | 21.76 | 21.13 | 29.92 | 28.27 | 19.46 | 30.75 | **31.75** |
| | PSP@10 | 6.85 | 6.73 | 9.24 | 9.00 | 13.2 | 12.96 | 8.81 | 13.22 | **13.76** |
| **ML-1M** | AUC | 0.8332 | 0.8330 | 0.9065 | 0.9045 | 0.9289 | 0.9069 | 0.8832 | **0.9457** | 0.9415 |
| | HR@10 | 13.07 | 12.93 | 24.63 | 23.25 | 27.43 | 28.54 | 21.7 | **31.51** | 31.16 |
| | HR@100 | 30.38 | 29.63 | 53.26 | 51.42 | 55.61 | 57.28 | 52.29 | **60.05** | 58.28 |
| | NDCG@10 | 13.84 | 13.74 | 25.65 | 24.44 | 28.85 | 29.88 | 22.14 | **32.82** | 32.52 |
| | NDCG@100 | 19.49 | 19.13 | 35.62 | 33.93 | 38.29 | 40.16 | 33.82 | **42.53** | 41.29 |
| | PSP@10 | 1.10 | 1.07 | 2.41 | 2.26 | 2.72 | 3.06 | 2.42 | **3.22** | 3.15 |
| **Douban** | AUC | 0.8945 | 0.8932 | 0.9288 | 0.9258 | 0.9391 | 0.8570 | 0.9129 | **0.9523** | 0.9510 |
| | HR@10 | 11.06 | 10.71 | 12.69 | 12.79 | 15.98 | 17.93 | 15.36 | **23.56** | 22.98 |
| | HR@100 | 17.07 | 16.63 | 20.29 | 19.69 | 22.38 | 25.41 | 22.82 | **28.37** | 27.20 |
| | NDCG@10 | 11.63 | 11.24 | 13.21 | 13.33 | 16.68 | 19.48 | 16.17 | **24.94** | 24.20 |
| | NDCG@100 | 12.63 | 12.27 | 14.96 | 14.39 | 17.20 | 19.55 | 17.32 | **23.26** | 22.21 |
| | PSP@10 | 0.52 | 0.50 | 0.63 | 0.63 | 0.86 | 1.06 | 0.87 | **1.28** | 1.24 |
| **Netflix** | AUC | 0.9249 | 0.9234 | 0.9234 | 0.9244 | | 0.9418 | 0.9495 | 0.9663* | **0.9728** |
| | HR@10 | 12.14 | 11.49 | 11.69 | 11.06 | | 26.03 | 20.6 | **29.69*** | 29.57 |
| | HR@100 | 28.47 | 27.66 | 27.72 | 26.76 | Timed | 50.35 | 44.53 | **50.88*** | 49.24 |
| | NDCG@10 | 12.34 | 11.72 | 12.04 | 11.48 | Out | 26.83 | 20.85 | **30.59*** | 30.54 |
| | NDCG@100 | 17.79 | 16.95 | 17.17 | 16.40 | | 35.09 | 29.22 | **36.59*** | 35.58 |
| | PSP@10 | 1.45 | 1.28 | 1.31 | 1.21 | | 3.59 | 2.77 | **3.75*** | 3.62 |

## 5  Experiments

**Setup.**  We use four recommendation datasets with varying sizes and sparsity characteristics. A brief set of data statistics can be found in Appendix B.3, Table 2. For each user in the dataset, we randomly split their interaction history into $80/10/10\%$ train-test-validation splits. Following recent warnings against unrealistic dense preprocessing of recommender system datasets [55, 57], we only prune users that have fewer than 3 interactions to enforce at least one interaction per user in the train, test, and validation sets. No such preprocessing is followed for items.

**Competitor methods & evaluation metrics.**  We compare ∞-AE with various baseline and SoTA competitors as surveyed in recent comparative analyses [1, 13]. More details on their architectures can be found in Appendix B.1. We evaluate all models on a variety of pertinent ranking metrics, namely AUC, HitRate (HR@k), Normalized Discounted Cumulative Gain (nDCG@k), and Propensity-scored Precision (PSP@k), each focusing on different components of the algorithm performance. A notable addition to our list of metrics compared to the literature is the PSP metric [22], which we adapt to the recommendation use case as an indicator of performance on tail items. The exact definition of all of these metrics can be found in Appendix B.5.

**Training details.**  We implement both ∞-AE and DISTILL-CF using JAX [8] along with the Neural-Tangents package [44] for the relevant NTK computations.[1,2] We re-use the official implementation of LightGCN, and implement the remaining competitors ourselves. To ensure a fair comparison, we conduct a hyper-parameter search for all competitors on the validation set. More details on the hyper-parameters for ∞-AE, DISTILL-CF, and all competitors can be found in Appendix B.3, Table 3. All of our experiments are performed on a single RTX 3090 GPU, with a random-seed initialization of 42. Additional training details about DISTILL-CF can be found in Appendix B.4.

---

[1]Our implementation for ∞-AE is available at https://github.com/noveens/infinite_ae_cf
[2]Our implementation for DISTILL-CF is available at https://github.com/noveens/distill_cf

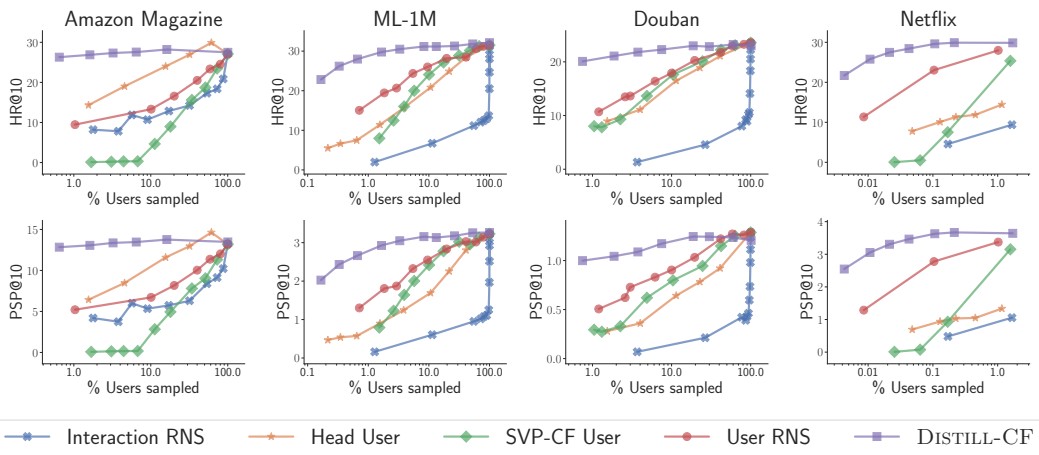

Figure 1: Performance of $\infty$-AE with the amount of users (log-scale) sampled according to different sampling strategies over the HR@10 and PSP@10 metrics. Results for the Netflix dataset have been clipped due to memory constraints. Results for the remaining metrics can be found in Appendix B.6, Figure 13.

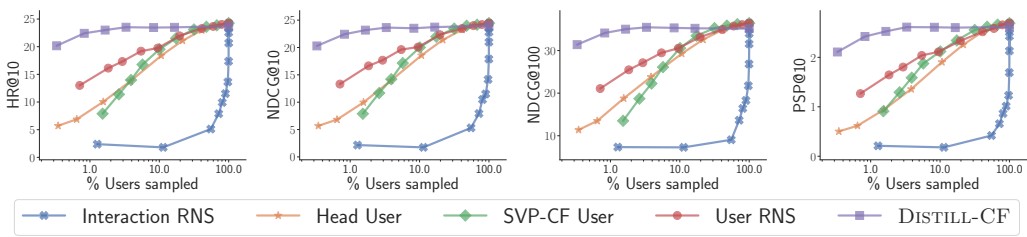

Figure 2: Performance of the EASE model trained on different amounts of users (log-scale) sampled by different samplers on the ML-1M dataset.

**Does $\infty$-AE outperform existing methods?** We compare the performance of $\infty$-AE with various baseline and competitor methods in Table 1. We also include the results of training $\infty$-AE on data synthesized by DISTILL-CF with an additional constraint of having a budget of only $\mu = 500$ synthetic users. For the sake of reference, for our largest dataset (Netflix), this equates to a mere 0.1% of the total users. There are a few prominent observations from the results in Table 1. First, $\infty$-AE significantly outperforms SoTA recommendation algorithms despite having only a single fully-connected layer, and also being much simpler to train and implement. Second, we note that $\infty$-AE trained on just 500 users generated by DISTILL-CF is able to attain $96 - 105\%$ of $\infty$-AE's performance on the full dataset while also outperforming all competitors trained on the *full* dataset.

**How sample efficient is $\infty$-AE?** Having noted from Table 1 that $\infty$-AE is able to outperform all SoTA competitors with as little as 0.1% of the original users, we now aim to better understand $\infty$-AE's sample complexity, *i.e.*, the amount of training data $\infty$-AE needs in order to perform accurate recommendation. In addition to DISTILL-CF, we use the following popular heuristics for down-sampling (more details in Appendix B.2): interaction random negative sampling (RNS); user RNS; head user sampling; and a coreset construction technique, SVP-CF user [57]. We then train $\infty$-AE on sampled data for different sampling budgets, while evaluating on the original test-set. We plot the performance for all datasets computed over the HR@10 and PSP@10 metrics in Figure 1. We observe that while all heuristic sampling strategies tend to be closely bound to the identity line with a slight preference to user RNS, $\infty$-AE when trained on data synthesized by DISTILL-CF tends to quickly saturate in terms of performance when the user budget is increased, even on the log-scale. This indicates DISTILL-CF's superiority in generating terse data summaries for $\infty$-AE, thereby allowing it to get SoTA performance on the largest datasets with as little as 500 users.

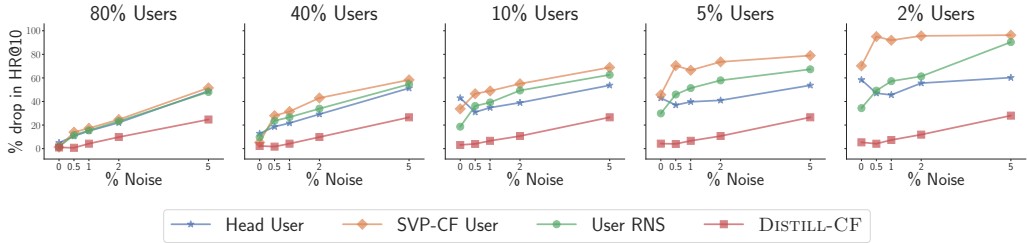

Figure 3: Performance drop of the EASE model trained on data sampled by different sampling strategies when there is varying levels of noise in the data. Performance drop is relative to training on the full, noise-free ML-1M dataset. Results for the remaining metrics can be found in Appendix B.6, Figure 14.

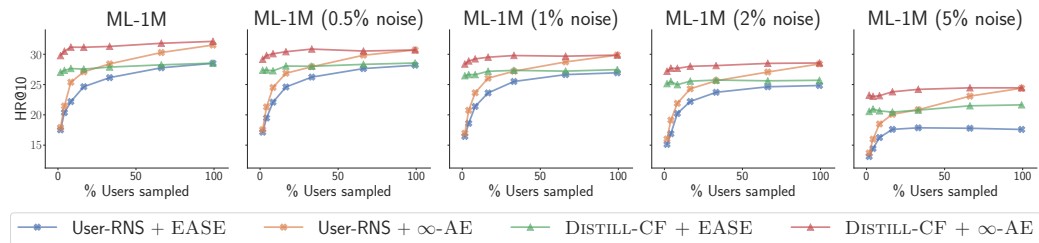

Figure 4: Performance of $\infty$-AE on data sampled by DISTILL-CF and User-RNS when there is noise in the data. Results for EASE have been added for reference. All results are on the ML-1M dataset. Results for the remaining metrics can be found in Appendix B.6, Figure 15.

**How transferable are the data summaries synthesized by DISTILL-CF?** In order to best evaluate the quality and universality of data summaries synthesized by DISTILL-CF, we train and evaluate EASE [59] on data synthesized by DISTILL-CF. Note that the inner loop of DISTILL-CF still consists of $\infty$-AE, but we nevertheless train and evaluate EASE to test the synthesized data's universality. We re-use the heuristic sampling strategies from the previous experiment for comparison with DISTILL-CF. From the results in Figure 2, we observe similar scaling laws as $\infty$-AE's for the heuristic samplers as well as DISTILL-CF. The semantically similar results for MVAE [32] are presented in Appendix B.6, Figure 10 for completeness. This behaviour validates the re-usability of data summaries generated by DISTILL-CF, because they transfer well to SoTA finite-width models, which were not involved in DISTILL-CF's user synthesis optimization.

**How robust are DISTILL-CF and $\infty$-AE to noise?** User feedback data is often noisy due to various biases (see [10] for a detailed review). Furthermore, due to the significant number of logged interactions in these datasets, recommender systems are often trained on down-sampled data in practice. Despite this, to the best of our knowledge, there is no prior work that explicitly studies the interplay between noise in the data and how sampling it affects downstream model performance. Consequently, we simulate a simple experiment: we add $x\%$ noise in the original train-set $\rightarrow$ sample the noisy training data $\rightarrow$ train recommendation models on the sampled data $\rightarrow$ evaluate their performance on the original, noise-free test-set. For the noise model, we randomly flip $x\%$ of the total number of items in the corpus for each user. In Figure 3, we compare the drop in HR@10 the EASE model suffers for different sampling strategies when different levels of noise are added to the MovieLens-1M dataset [15]. We make a few main observations: (1) unsurprisingly, sampling noisy data compounds the performance losses of learning algorithms; (2) DISTILL-CF has the best noise:sampling:performance trade-off compared to other sampling strategies, with an increasing performance gap relative to other samplers as we inject more noise into the original data; and (3) as we down-sample noisy data more aggressively, head user sampling improves relative to other samplers, simply because these head users are the least affected by our noise injection procedure.

Furthermore, to better understand $\infty$-AE's denoising capabilities, we repeat the aforementioned noise-injection experiment but now train $\infty$-AE on down-sampled, noisy data. In Figure 4, we track

the change in $\infty$-AE's performance as a function of the number of users sampled, and the amount of noise injected before sampling. We also add the semantically equivalent results for the EASE model for reference. Firstly, we note that the full-data performance-gap between $\infty$-AE and EASE significantly increases when there is more noise in the data, demonstrating $\infty$-AE's robustness to noise, even when its not specifically optimized for it. Furthermore, looking at the 5% noise injection scenario, we notice two counter-intuitive observations: (1) training EASE on tiny data summaries synthesized by DISTILL-CF is better than training it on the full data; and (2) solely looking at data synthesized by DISTILL-CF for EASE, we notice the best performance when we have a lower user sampling budget. Both of these observations call for more investigation of a data-centric viewpoint to recommendation, *i.e.*, focusing more on the quality of collected data rather than its quantity.

**Applications to continual learning.** Continual learning (see [45] for a detailed review) is an important area for recommender systems, because these systems are typically updated at regular intervals. A continual learning scenario involves data that is split into multiple periods, with the predictive task being: given data until the $i^{\text{th}}$ period, maximize algorithm performance for prediction on the $(i + 1)^{\text{th}}$ period. ADER [35] is a SoTA continual learning model for recommender systems, that injects the most informative user sequences from the last period to combat the catastrophic forgetting problem [52]. An intuitive application for DISTILL-CF is to synthesize succinct data summaries of the last period and inject these instead. To compare these approaches, we simulate a continual learning scenario by splitting the MovieLens-1M dataset into 17 equal sized epochs,

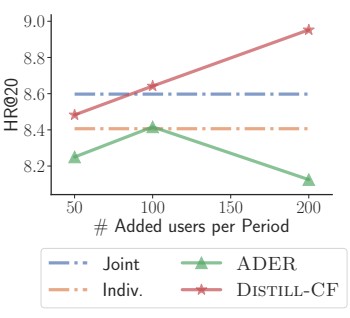

Figure 5: DISTILL-CF for continual learning.

and perform experiments with MVAE [32] for each period. Note that in DISTILL-CF, we still use $\infty$-AE to synthesize data summaries (inner loop). We also compare with two baselines: (1) Joint: concatenate the data from all periods before the current; and (2) Individual: use the data only from the current period. As we can see from Figure 5, DISTILL-CF consistently outperforms ADER and the baselines, again demonstrating its ability to generate high-fidelity data summaries.

## 6 Conclusion & Future Work

In this work, we proposed two complementary ideas: $\infty$-AE, an infinite-width autoencoder for modeling recommendation data, and DISTILL-CF for creating tiny, high-fidelity data summaries of massive datasets for subsequent model training. To our knowledge, our work is the first to employ and demonstrate that infinite-width neural networks can beat complicated SoTA models on recommendation tasks. Further, the data summaries synthesized through DISTILL-CF outperform generic samplers and demonstrate further performance gains for $\infty$-AE as well as finite-width SoTA models despite being trained on orders of magnitude less data.

Both our proposed methods are closely linked with one another: $\infty$-AE's closed-loop formulation is especially crucial in the practicality of DISTILL-CF, whereas DISTILL-CF's ability to distill the entire dataset's knowledge into small summaries helps $\infty$-AE to scale to large datasets. Moreover, the Gumbel sampling trick enables us to adapt data distillation techniques designed for continuous, real-valued, dense domains to heterogeneous, semi-structured, and sparse domains like recommender systems and graphs. We additionally explore the strong denoising effect observed with DISTILL-CF, noting that in the case of noisy data, models trained on considerably less data synthesized by DISTILL-CF perform better than the same model trained on the entire original dataset. These observations lead us to contemplate a much larger, looming question: *Is more data what you need for recommendation?* Our results call for further investigation on the data-centric viewpoint of recommendation.

The findings of our paper open up numerous promising research directions. First, building such closed-form, easy-to-implement infinite networks is beneficial for various downstream practical applications like search, sequential recommendation, or CTR prediction. Further, the anonymization achieved by synthesizing fake data summaries is crucial for mitigating the privacy risks associated with confidential or PII datasets. Another direction is analyzing the environmental impact and reduction in carbon footprint as our experiments show that models can achieve similar performance gains when trained on much less data.

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
