# A Appendix: Pseudo-codes

---

**Algorithm 1** $\infty$-AE model training

---

**Input:** User set $\mathcal{U}$ ; dataset $\mathbf{X} \in \mathbb{R}^{|\mathcal{U}| \times |\mathcal{I}|}$; NTK $\mathbb{K} : \mathbb{R}^{|\mathcal{I}|} \times \mathbb{R}^{|\mathcal{I}|} \mapsto \mathbb{R}$ ; regularization const. $\lambda \in \mathbb{R}$
**Output:** Dual parameters $\alpha \in \mathbb{R}^{|\mathcal{U}| \times |\mathcal{I}|}$

1: **procedure** FIT($\mathcal{U}, \mathbf{X}, \mathbb{K}$)
2:     $\mathbf{K} \leftarrow [0]_{|\mathcal{U}| \times |\mathcal{U}|}$         $\triangleright$ Zero Initialization
3:     $\mathbf{K}_{u,v} \leftarrow \mathbb{K}(\mathbf{X}_u, \mathbf{X}_v) \quad \forall u \in \mathcal{U}, v \in \mathcal{U}$
4:     $\alpha \leftarrow (\mathbf{K} + \lambda I)^{-1} \cdot \mathbf{X}$
5:     **return** $\alpha$

---

**Algorithm 2** $\infty$-AE inference

---

**Input:** User set $\mathcal{U}$ ; dataset $\mathbf{X} \in \mathbb{R}^{|\mathcal{U}| \times |\mathcal{I}|}$; NTK $\mathbb{K} : \mathbb{R}^{|\mathcal{I}|} \times \mathbb{R}^{|\mathcal{I}|} \mapsto \mathbb{R}$; dual params. $\alpha \in \mathbb{R}^{|\mathcal{U}| \times |\mathcal{I}|}$ ; inference user history $\hat{\mathbf{X}}_u \in \mathbb{R}^{|\mathcal{I}|}$
**Output:** Prediction $\hat{y} \in \mathbb{R}^{|\mathcal{I}|}$

1: **procedure** PREDICT($\mathcal{U}, \mathbf{X}, \hat{\mathbf{X}}_u, \mathbb{K}, \alpha$)
2:     $\mathbf{K} \leftarrow [0]_{|\mathcal{U}|}$         $\triangleright$ Zero Initialization
3:     $\mathbf{K}_v \leftarrow \mathbb{K}(\hat{\mathbf{X}}_u, \mathbf{X}_v) \quad \forall v \in \mathcal{U}$
4:     $\hat{y} \leftarrow softmax(\mathbf{K} \cdot \alpha)$
5:     **return** $\hat{y}$

---

**Algorithm 3** Data synthesis using DISTILL-CF

---

**Input:** User set $\mathcal{U}$ ; dataset $\mathbf{X} \in \mathbb{R}^{|\mathcal{U}| \times |\mathcal{I}|}$; NTK $\mathbb{K} : \mathbb{R}^{|\mathcal{I}|} \times \mathbb{R}^{|\mathcal{I}|} \mapsto \mathbb{R}$ ; support user size $\mu \in \mathbb{R}$ ; gumbel softmax temperature $\tau \in \mathbb{R}$ ; reg. const. $\lambda_2 \in \mathbb{R}$ ; SGD batch-size $b$, step-size $\eta \in \mathbb{R}$
**Output:** Synthesized data $\mathbf{X}^{\mathbf{s}} \in \mathbb{R}^{\mu \times |\mathcal{I}|}$

1: **procedure** SAMPLE($n, \mathcal{U}, \mathbf{X}$)
2:     $\mathcal{U}' \sim \mathcal{U}$                                      $\triangleright$ Randomly sample $n$ users from $\mathcal{U}$
3:     $\mathbf{X}' \leftarrow \mathbf{X_u} \quad \forall u \in \mathcal{U}'$                      $\triangleright$ Retrieve corresponding rows from $\mathbf{X}$
4:     **return** $\mathcal{U}', \mathbf{X}'$
5: **procedure** SYNTHESIZE($\mathcal{U}, \mathbf{X}, \mathbb{K}$)
6:     $\mathcal{U}^s, \mathbf{X}^{\mathbf{s}} \leftarrow$ SAMPLE($\mu, \mathcal{U}, \mathbf{X}$)                      $\triangleright$ Sample support data
7:     **for** $steps = 0 \dots \xi$ **do**
8:         $\hat{\mathbf{X}}^{\mathbf{s}} \leftarrow \sigma \left[ \sum_{i=1}^{\gamma} gumbel_\tau(softmax(\mathbf{X}^{\mathbf{s}})) \right]$
9:         $\alpha^s \leftarrow$ FIT($\mathcal{U}^s, \hat{\mathbf{X}}^{\mathbf{s}}, \mathbb{K}$)                      $\triangleright$ Fit $\infty$-AE on support data
10:        $\mathcal{U}^b, \mathbf{X}^{\mathbf{b}} \leftarrow$ SAMPLE($b, \mathcal{U}, \mathbf{X}$)
11:        $\tilde{\mathbf{X}} \leftarrow [0]_{b \times |\mathcal{I}|}$
12:        $\tilde{\mathbf{X}_\mathbf{u}} \leftarrow$ PREDICT($\mathcal{U}^s, \hat{\mathbf{X}}^{\mathbf{s}}, \mathbf{X_u^b}, \mathbb{K}, \alpha^s$) $\quad \forall u \sim \mathcal{U}^b$       $\triangleright$ Predict for all sampled users
13:        $L \leftarrow \mathbf{X}^{\mathbf{b}} \cdot log(\tilde{\mathbf{X}}) + (1 - \mathbf{X}^{\mathbf{b}}) \cdot log(1 - \tilde{\mathbf{X}}) + \lambda_2 \cdot ||\hat{\mathbf{X}}^{\mathbf{s}}||_1$       $\triangleright$ Re-construction loss
14:        $\mathbf{X}^{\mathbf{s}} \leftarrow \mathbf{X}^{\mathbf{s}} - \eta \cdot \frac{\partial L}{\partial \mathbf{X}^{\mathbf{s}}}$                      $\triangleright$ SGD on $\mathbf{X}^{\mathbf{s}}$
15:    **return** $\mathbf{X}^{\mathbf{s}}$

---

# B Appendix: Experiments

## B.1 Baselines & Competitor Methods

We provide a high-level overview of the competitor models used in our experiments:

- **PopRec:** This implicit-feedback baseline simply recommends the most *popular* items in the catalog irrespective of the user. Popularity of an item is estimated by their empirical frequency in the logged train-set.

- **Bias-only:** This baseline learns scalar user and item biases for all users and item in the train-set, optimized by solving a least-squares regression problem between the predicted and observed relevance. More formally, given a user $u$ and an item $i$, the relevance is predicted as $\hat{r}_{u,i} = \alpha + \beta_u + \beta_i$, where $\alpha \in \mathbb{R}$ is a global offset bias, and $\beta_u, \beta_i \in \mathbb{R}$ are the user and item specific biases respectively. This model doesn't consider any cross user-item interactions, and hence lacks expressivity.

- **MF:** Building on top of the bias-only baseline, the Matrix Factorization algorithm tries to represent the users and items in a shared latent space, modeling their relevance by the dot-product of their representations. More formally, given a user $u$ and an item $i$, the relevance

Table 2: Brief set of statistics of the datasets used in this paper.

| Dataset | # Users | # Items | # Interactions | Sparsity |
|---|---|---|---|---|
| **Amazon Magazine [42]** | 3k | 1.3k | 12k | 99.7% |
| **ML-1M [15]** | 6k | 3.7k | 1M | 95.6% |
| **Douban [69]** | 2.6k | 34k | 1.2M | 98.7% |
| **Netflix [5]** | 476k | 17k | 100M | 98.9% |

is predicted as $\hat{r}_{u,i} = \alpha + \beta_u + \beta_i + (\gamma_u \cdot \gamma_i)$, where $\alpha, \beta_u, \beta_i$ are global, user, and item biases respectively, and $\gamma_u, \gamma_i \in \mathbb{R}^d$ represent the learned user and item representations. The biases and latent representations in this model are estimated by optimizing for the Bayesian Personalized Ranking (BPR) loss [49].

- **NeuMF [18]:** As a neural extension to MF, Neural Matrix Factorization aims to replace the linear cross-interaction between the user and item representations with an arbitrarily complex, non-linear neural network. More formally, given a user $u$ and an item $i$, the relevance is predicted as $\hat{r}_{u,i} = \alpha + \beta_u + \beta_i + \phi(\gamma_u, \gamma_i)$, where $\alpha, \beta_u, \beta_i$ are global, user, and item biases respectively, $\gamma_u, \gamma_i \in \mathbb{R}^d$ represent the learned user and item representations, and $\phi : \mathbb{R}^d \times \mathbb{R}^d \mapsto \mathbb{R}$ is a neural network. The parameters for this model are again optimized using the BPR loss.

- **MVAE [32]:** This method proposed using variational auto-encoders for the task of collaborative filtering. Their main contribution was to provide a principled approach to perform variational inference for the task of collaborative filtering.

- **LightGCN [17]:** This simplistic Graph Convolution Network (GCN) framework removes all the steps in a typical GCN [28], only keeping a linear neighbourhood aggregation step. This *light* model demonstrated promising results for the collaborative filtering scenario, despite its simple architecture. We use the official public implementation[3] for our experiments.

- **EASE [59]:** This linear model proposed doing ordinary least squares regression by estimating an item-item similarity matrix, that can be viewed as a zero-depth auto-encoder. Performing regression gives them the benefit of having a closed-form solution. Despite its simplicity, EASE has been shown to out-perform most of the deep non-linear neural networks for the task of collaborative filtering.

## B.2 Sampling strategies

Given a recommendation dataset $\mathcal{D} := \{(\text{user}_i, \text{item}_i, \text{relevance}_i)\}_{i=1}^n$ consisting of $n$ user-item interactions defined over a set of users $\mathcal{U}$, set of items $\mathcal{I}$, and operating over a binary relevance score (implicit feedback); we aim to make a $p\%$ sub-sample of $\mathcal{D}$, defined in terms of number of interactions. Below are the different sampling strategies we used in comparison with DISTILL-CF:

- **Interaction-RNS:** Randomly sample $p\%$ interactions from $\mathcal{D}$.

- **User-RNS:** Randomly sample a user $u \sim \mathcal{U}$, and add all of its interactions into the sampled set. Keep repeating this procedure until the size of sampled set is less than $\frac{p \times n}{100}$.

- **Head user:** Sample the user $u$ from $\mathcal{U}$ with the most number of interactions, and add all of its interactions into the sampled set. Remove $u$ from $\mathcal{U}$. Keep repeating this procedure until the size of sampled set is less than $\frac{p \times n}{100}$.

- **SVP-CF user [57]:** This coreset mining technique proceeds by first training a proxy model on $\mathcal{D}$ for $e$ epochs. SVP-CF then modifies the forgetting events approach [61], and counts the inverse AUC for each user in $\mathcal{U}$, averaged over all $e$ epochs. Just like head-user sampling, we now iterate over users in the order of their forgetting count, and keep sampling users until we exceed our sampling budget of $\frac{p \times n}{100}$ interactions. We use the bias-only model as the proxy.

---

[3]https://github.com/gusye1234/LightGCN-PyTorch

Table 3: List of all the hyper-parameters grid-searched for $\infty$-AE, DISTILL-CF, and baselines.

| Hyper-Parameter | Model | Amz. Magazine | ML-1M | Douban | Netflix |
|---|---|---|---|---|---|
| Latent size | MF
NeuMF
LightGCN
MVAE | {4, 8, 16, 32, 50, 64, 128} | | | |
| # Layers | MF
NeuMF
LightGCN
MVAE | {1, 2, 3} | | | |
| | $\infty$-AE | {1} | | | |
| Learning rate | MF
NeuMF
LightGCN
MVAE | {0.001, 0.006, 0.01} | | | {0.006} |
| | DISTILL-CF | {0.04} | | | |
| Dropout | MF
NeuMF
LightGCN
MVAE | {0.0, 0.3, 0.5} | | | |
| $\lambda$ | EASE
$\infty$-AE
DISTILL-CF | {1, 10, 100, 1K, 10K}
{0.0, 1.0, 5.0, 20.0 50.0, 100.0}
{1e-5, 1e-3, 0.1, 1.0, 5.0, 50.0} | | | |
| $\lambda_2$ | DISTILL-CF | $\dfrac{\{\text{1e-3, 10.0}\}}{\text{avg. \# of interactions per user}}$ | | | |
| $\tau$ | DISTILL-CF | {0.3, 0.5, 0.7, 5.0} | | | |
| $\gamma$ | DISTILL-CF | {50, 100, 200} | {200, 500, 700} | {500, 1K, 2K} | {500, 700} |

## B.3    Data statistics & hyper-parameter configurations

We present a brief summary of statistics of the datasets used in our experiments in Table 2, and list all the hyper-parameter configurations tried for $\infty$-AE, DISTILL-CF, and other baselines in Table 3.

## B.4    Additional training details

We now discuss additional training details about DISTILL-CF that could not be presented in the main text due to space constraints. Firstly, we make use of a validation-set, and evaluate the performance of the $\infty$-AE model trained in DISTILL-CF's inner-loop to perform hyper-parameter search, as well as early exit. Note that we don't validate the inner-loop's $\lambda$ at every outer-loop iteration, but keep changing it on-the-fly at each validation cycle. We notice this trick gives us a much faster convergence compared to keeping $\lambda$ fixed for the entire distillation procedure, and validating for it like other hyper-parameters.

We also discuss the Gumbel sampling procedure described in Equation (3), in more detail. Given the sampling prior matrix $X^s$, that intuitively denotes the importance of sampling a specific user-item interaction, we intend to sample $\hat{X}^s$ which will finally be used for downstream model applications. Note that for each row (user) in $X^s$, we need multiple, but variable number of samples to conform to the Zipfian law for user and item popularity. This requirement in itself rejects the possibility to use top-K sampling which will sample the same number of items for each row. Furthermore, to keep $\hat{X}^s \sim X^s$ sampling part of the optimization procedure, we need to compute the gradients of the logistic objective in Equation (3) with respect to $X^s$, and hence need the entire process to be differentiable. This requirement prohibits the usage of other popular strategies like Nucleus sampling [19], which is non-differentiable. To workaround all the requirements, we devise a multi-step Gumbel sampling strategy where for each row (user) we take a fixed number of Gumbel samples ($\gamma$), with replacement, followed by taking a union of all the sampled user-item interactions. Note that the union

operation ensures that due to sampling with replacement, if a user-item interaction is sampled multiple times, we sample it only once. Hence, the number of sampled interactions is strictly upper-bounded by $\gamma \times |\mathcal{I}|$. To be precise, the sampling procedure is formalized below:

$$\hat{X}^s_{u,i} = \sigma \left[ \sum^{\gamma} \frac{exp(\frac{\log(X^s_{u,i}) + g_{u,i}}{\tau})}{\sum_{j \in \mathcal{I}} exp(\frac{\log(X^s_{u,j}) + g_{u,j}}{\tau})} \right] \quad \text{s.t.} \quad g_{u,i} \sim \text{Gumbel}(\mu = 0, \beta = 1) \quad \forall u \in \mathcal{U}, i \in \mathcal{I}$$

Where $\sigma$ represents an appropriate function which clamps all values between $0$ and $1$. In our experiments, we use hard-tanh.

### B.5 Evaluation metrics

We now present formal definitions of all the ranking metrics used in this study:

- **AUC:** Intuitively defined as a threshold independent classification performance measure, AUC can also be interpreted as the expected probability of a recommender system ranking a positive item over a negative item for any given user. More formally, given a user $u$ from the user set $\mathcal{U}$ with its set of positive interactions $\mathcal{I}^+_u \subseteq \mathcal{I}$ with a similarly defined set of negative interactions $\mathcal{I}^-_u = \mathcal{I} \backslash \mathcal{I}^+_u$, the AUC for a relevance predictor $\phi(u, i)$ is defined as:

$$\text{AUC}(\phi) := \underset{u \sim \mathcal{U}}{\mathbb{E}} \left[ \underset{i^+ \sim \mathcal{I}^+_u}{\mathbb{E}} \left[ \underset{i^- \sim \mathcal{I}^-_u}{\mathbb{E}} \left[ \phi(u, i^+) > \phi(u, i^-) \right] \right] \right]$$

- **HitRate (HR@k):** Another name for the recall metric, this metric estimates how many positive items are predicted in a top-k recommendation list. More formally, given recommendation lists $\hat{Y}_u \subseteq \mathcal{I}^K \quad \forall u \in \mathcal{U}$ and the set of positive interactions $\mathcal{I}^+_u \subseteq \mathcal{I} \quad \forall u \in \mathcal{U}$:

$$\text{HR@k} := \underset{u \sim \mathcal{U}}{\mathbb{E}} \left[ \frac{|\hat{Y}_u \cap \mathcal{I}^+_u|}{|\mathcal{I}^+_u|} \right]$$

- **Normalized Discounted Cumulative Gain (nDCG@k):** Unlike HR@k which gives equal importance to all items in the recommendation list, the nDCG@k metric instead gives a higher importance to items predicted higher in the recommendation list and performs logarithmic discounting further down. More formally, given *sorted* recommendation lists $\hat{Y}_u \subseteq \mathcal{I}^K \quad \forall u \in \mathcal{U}$ and the set of positive interactions $\mathcal{I}^+_u \subseteq \mathcal{I} \quad \forall u \in \mathcal{U}$:

$$\text{nDCG@k} := \underset{u \sim \mathcal{U}}{\mathbb{E}} \left[ \frac{\text{DCG}_u}{\text{IDCG}_u} \right] \ ; \ \text{DCG}_u := \sum_{i=1}^{k} \frac{\hat{Y}^i_u \in \mathcal{I}^+_u}{log_2(i+1)} \ ; \ \text{IDCG}_u := \sum_{i=1}^{|\mathcal{I}^+_u|} \frac{1}{log_2(i+1)}$$

- **Propensity-scored Precision (PSP@k):** Originally introduced in [22] for extreme classification scenarios [46, 21, 37], the PSP@k metric intuitively accounts for missing labels (items in the case of recommendation) by dividing the true relevance of an item (binary) with a propensity correction term. More formally, given recommendation lists $\hat{Y}_u \subseteq \mathcal{I}^K \quad \forall u \in \mathcal{U}$, the set of positive interactions $\mathcal{I}^+_u \subseteq \mathcal{I} \quad \forall u \in \mathcal{U}$, and a propensity model $\phi : \mathcal{I} \mapsto \mathbb{R}$:

$$\text{PSP@k} := \underset{u \sim \mathcal{U}}{\mathbb{E}} \left[ \frac{\text{uPSP}_u}{\text{mPSP}_u} \right] \ ; \ \text{uPSP}_u := \frac{1}{k} \cdot \sum_{i=1}^{k} \frac{\hat{Y}^i_u \in \mathcal{I}^+_u}{\phi(\hat{Y}^i_u)} \ ; \ \text{mPSP}_u := \sum_{i \in \mathcal{I}^+_u} \frac{1}{\phi(i)}$$

For $\phi$, we adapt the propensity model proposed in [22] for the case of recommendation as:

$$\phi(i) \equiv \underset{u \sim \mathcal{U}}{\mathbb{E}} \left[ P(r_{u,i} = 1 | r^*_{u,i} = 1) \right] = \frac{1}{1 + C \cdot e^{-A \cdot log(n_i + B)}} \ ; \ C = (logN - 1) \cdot (B+1)^A$$

Where, $N$ represents the total number of interactions in the dataset, and $n_i$ represents the empirical frequency of item $i$ in the dataset. We use $A = 0.55$ and $B = 1.5$ for our experiments.

## B.6 Additional experiments

**How does depth affect $\infty$-AE?** To better understand the effect of depth on an infinitely-wide auto-encoder's performance for recommendation, we extend $\infty$-AE to multiple layers and note its downstream performance change in Figure 6. The prominent observation is that models tend to get *worse* as they get deeper, with generally good performance in the range of $1 - 2$ layers, which also has been common practical knowledge even for finite-width recommender systems.

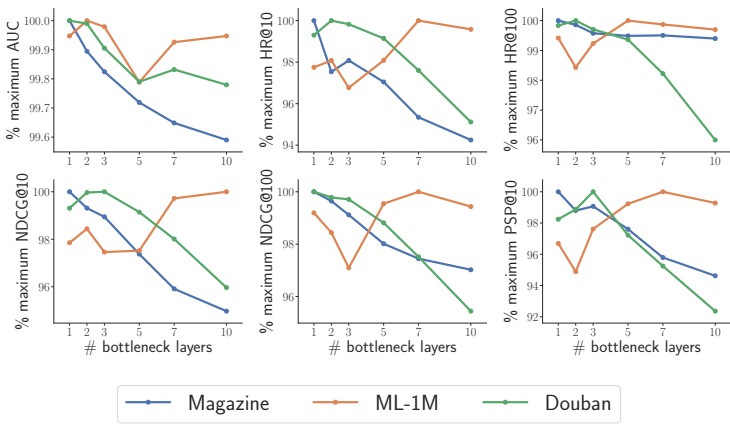

Figure 6: Performance of $\infty$-AE with varying depths. The y-axis represents the normalized metric *i.e.* performance relative to the best depth for a given metric.

**How does $\infty$-AE perform on cold users & cold items?** Cold-start has been one of the hardest problems in recommender systems — how to best model users or items that have very little training data available? Even though $\infty$-AE doesn't have any extra modeling for these scenarios, we try to better understand the performance of $\infty$-AE over users' and items' coldness spectrum. In Figure 7, we quantize different users and items based on their coldness (computed by their empirical occurrence in the train-set) into equisized buckets and measure different models' performance only on the binned users or items. We note $\infty$-AE's dominance over other competitors especially over the tail, head-users; and torso, head-items.

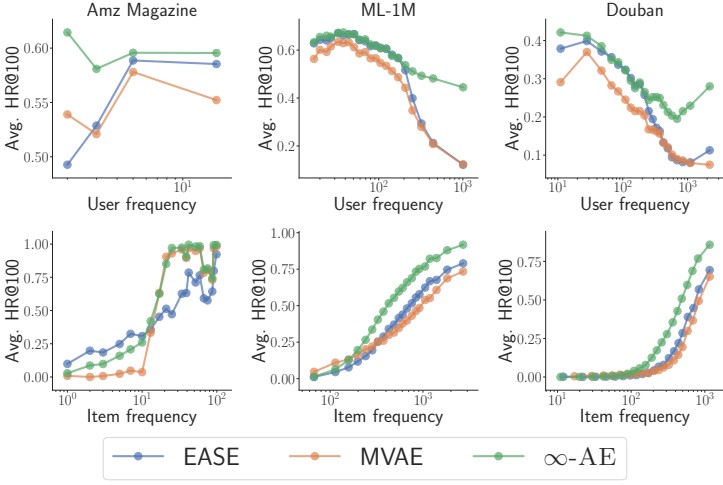

Figure 7: Performance comparison of $\infty$-AE with SoTA finite-width models stratified over the coldness of users and items. The y-axis represents the average HR@100 for users/items in a particular quanta. All user/item bins are equisized.

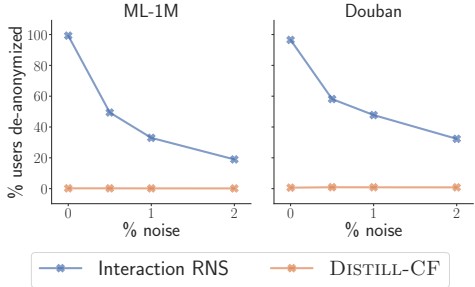
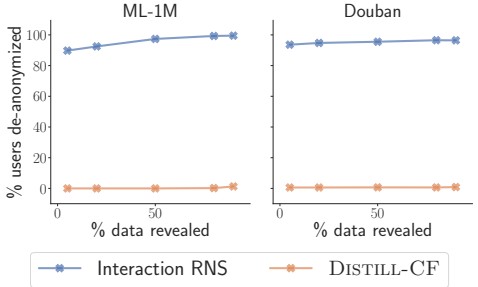

Figure 8: Amount of noise added in $\mathcal{D}'$ *vs.* % of users de-anonymized.

Figure 9: Amount of data revealed in $\mathcal{D}'$ *vs.* % of users de-anonymized.

**How anonymized is the data synthesized by DISTILL-CF?** Having evaluated the fidelity of distills generated using DISTILL-CF, we now focus on understanding its anonymity and syntheticity. For the generic data down-sampling case, the algorithm presented in [38] works well to de-anonymize the Netflix prize dataset. The algorithm assumes a complete, non-PII dataset $\mathcal{D}$ along with an incomplete, noisy version of the same dataset $\mathcal{D}'$, but also has the sensitive PII available. We simulate a similar setup but extend to datasets other than Netflix, by following a simple down-sampling and noise addition procedure: given a sampling strategy $s$, down-sample $\mathcal{D}$ and add $x\%$ noise by randomly flipping $x\%$ of the total items for each user to generate $\mathcal{D}'$. We then use our implementation of the algorithm proposed in [38] to *match* the corresponding users in the original, noise-free dataset $\mathcal{D}$.

However, if instead of a down-sampled dataset $\mathcal{D}'$, a data distill of $\mathcal{D}$ (using DISTILL-CF), let's say $\tilde{\mathcal{D}}$, is made publicly available. The task of de-anonymization can no longer be carried out by simply *matching* user histories from $\mathcal{D}'$ to $\mathcal{D}$, since $\mathcal{D}$ is no longer available. The only solution now is to *predict* the missing items in $\mathcal{D}'$. Note that this this task is easier than the usual recommendation problem, as the user histories to complete in $\mathcal{D}'$ do exist in some incoherent way in the data distill $\tilde{\mathcal{D}}$, and is more similar to train-set prediction. To test this out, we formulate a simple experiment: given a data distill $\tilde{\mathcal{D}}$, an incomplete, noisy subset $\mathcal{D}'$ with PII information, and also hypothetically the number of missing items for each user in $\mathcal{D}'$ — how accurately can we predict the *exact* set of missing items in $\mathcal{D}'$ using an $\infty$-AE model trained on $\tilde{\mathcal{D}}$.

We perform experiments for both the cases of data-sampling and data-distillation. In Figure 8, we measure the % of users de-anonymized using the aforementioned procedures. We interestingly note no level of de-anonymization with the data-distill, even if there's no noise in $\mathcal{D}'$. We also note the expected observation for the data-sampling case: less users are de-anonymized when there's more noise in $\mathcal{D}'$. In Figure 9, we now control the amount of data revealed in $\mathcal{D}'$. We again note the same observation: even with 90% of the correct data from $\mathcal{D}$ revealed in $\mathcal{D}'$ with 0% of noise, we still note a very tiny 0.86% of user de-anonymization with data-distillation, whereas 96.43% with data-sampling for the Douban dataset.

**How does DISTILL-CF compare to data augmentation approaches?** We compare the quality of data synthesized by DISTILL-CF with generative models proposed for data augmentation. One such SoTA method is AR-CF [9], which leverages two conditional GANs [36] to generate fake users and fake items. For our experiment, we focus only on AR-CF's user generation sub-network and consequently train the EASE [59] model *only* on these synthesized users, while testing on the original test-set for the MovieLens-1M dataset. We plot the results in Figure 11, comparing the amount of users synthesized according to different strategies and plot the HR@10 of the correspondingly trained model. The results signify that training models only on data synthesized by data augmentation models is impractical, as these users have only been optimized for being *realistic*, whereas the users synthesized by DISTILL-CF are optimized to be *informative for model training*. The same observation tends to hold true for the case of images as well [67].

**Additional experiments on the generality of data summaries synthesized by DISTILL-CF.** Continuing the results in Figure 2, we now train and evaluate MVAE [32] on data synthesized by DISTILL-CF. Note that the inner loop of DISTILL-CF still consists of $\infty$-AE, but we nevertheless

train and evaluate MVAE to test the synthesized data's universality. We re-use the heuristic sampling strategies from Figure 2 for comparison with DISTILL-CF. From the results in Figure 10, we observe similar scaling laws as EASE for the heuristic samplers as well as DISTILL-CF. A notable exception is interaction RNS, that scales like ∞-AE. Another interesting observation to note is that training MVAE on the full data performs slightly worse than training MVAE on the same amount of data synthesized by DISTILL-CF. This behaviour validates the re-usability of data summaries generated by DISTILL-CF, because they transfer well to SoTA finite-width models, which were not involved in DISTILL-CF's user synthesis procedure.

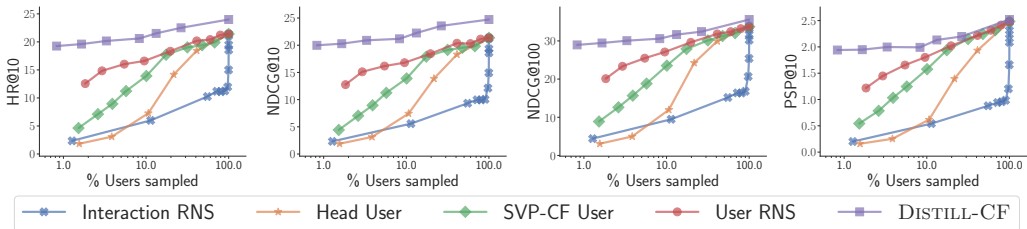

Figure 10: Performance of the MVAE model trained on different amounts of users (log-scale) sampled by different samplers on the ML-1M dataset.

**Additional experiments on Continual learning.** Continuing the results in Figure 5, we plot the results stratified per period for the MovieLens-1M dataset in Figure 12. The results are a little noisy, but we can observe that exemplar data distilled with DISTILL-CF has better performance for a majority of the data periods. Note that we use the official public implementation[4] of ADER.

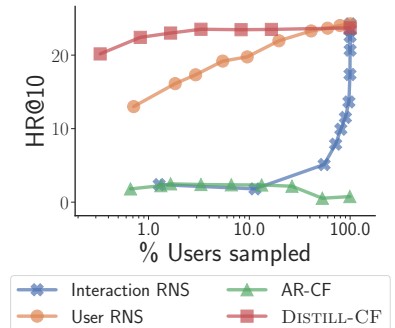

Figure 11: Performance of EASE on varying amounts of data sampled/synthesized using various strategies for the MovieLens-1M dataset.

Figure 12: Per-period evaluation of the MVAE model on various continual learning strategies as discussed in Section 5.

**Additional plots for the sample complexity of ∞-AE.** In addition to Figure 1 in the main paper, we visualize the sample complexity of ∞-AE for all datasets and all metrics in Figure 13. We notice similar trends for all metrics across datasets.

**Additional plots on the robustness of DISTILL-CF & ∞-AE to noise.** In addition to Figure 3 and Figure 4 in the main paper, we plot results for the EASE model trained on data sampled by different sampling strategies, when there's varying levels of noise in the original data. We plot this for the MovieLens-1M dataset and all metrics in Figure 13. We notice similar trends for all metrics across datasets. We also plot the sample complexity results for EASE and ∞-AE over the MovieLens-1M dataset and all metrics in Figure 15. We observe similar trends across metrics.

---

[4]https://github.com/doublemul/ADER available with the MIT license.

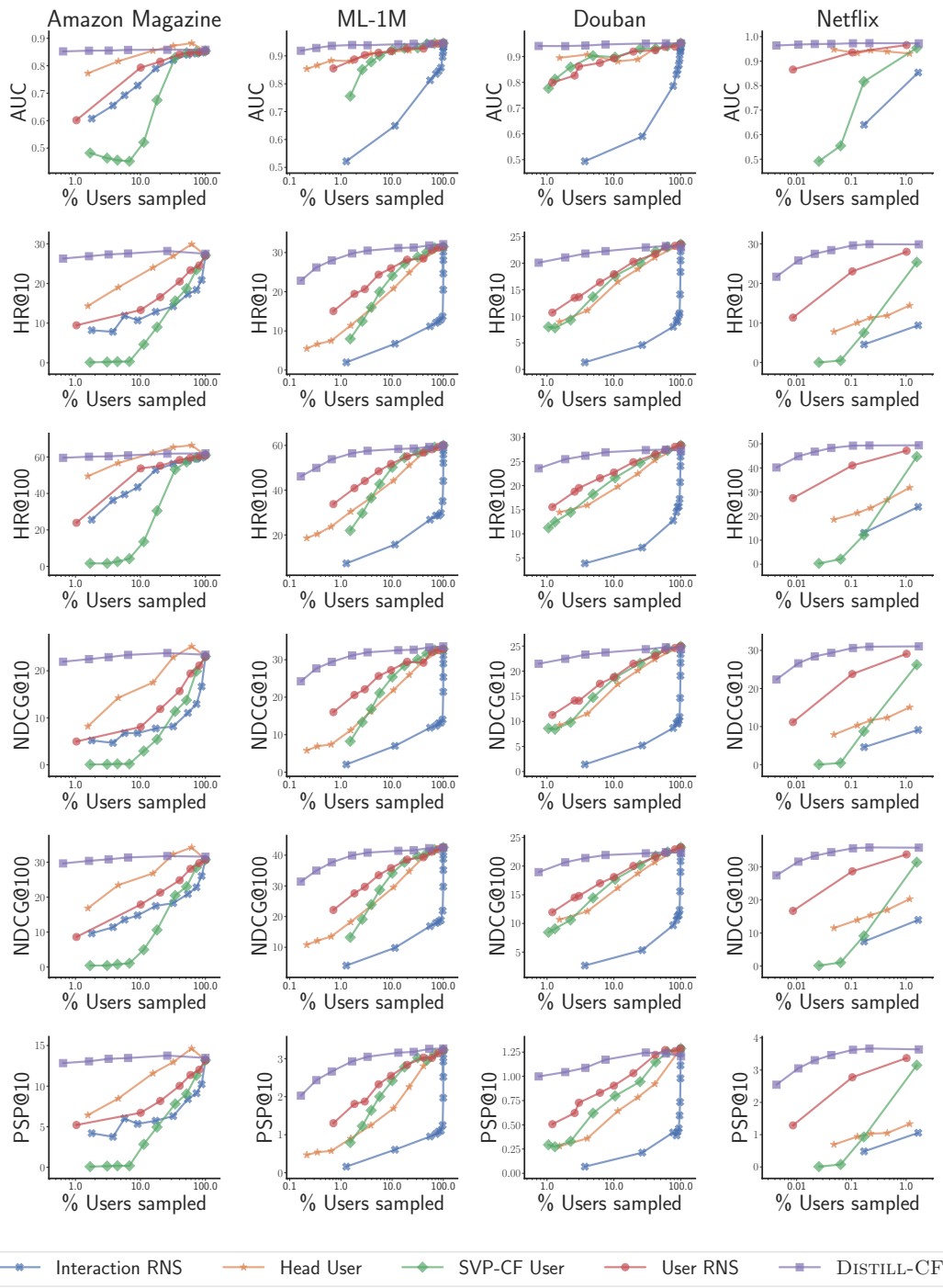

Figure 13: Performance of $\infty$-AE with the amount of users sampled according to different sampling strategies over different metrics. Each column represents a single dataset, and each row represents an evaluation metric.

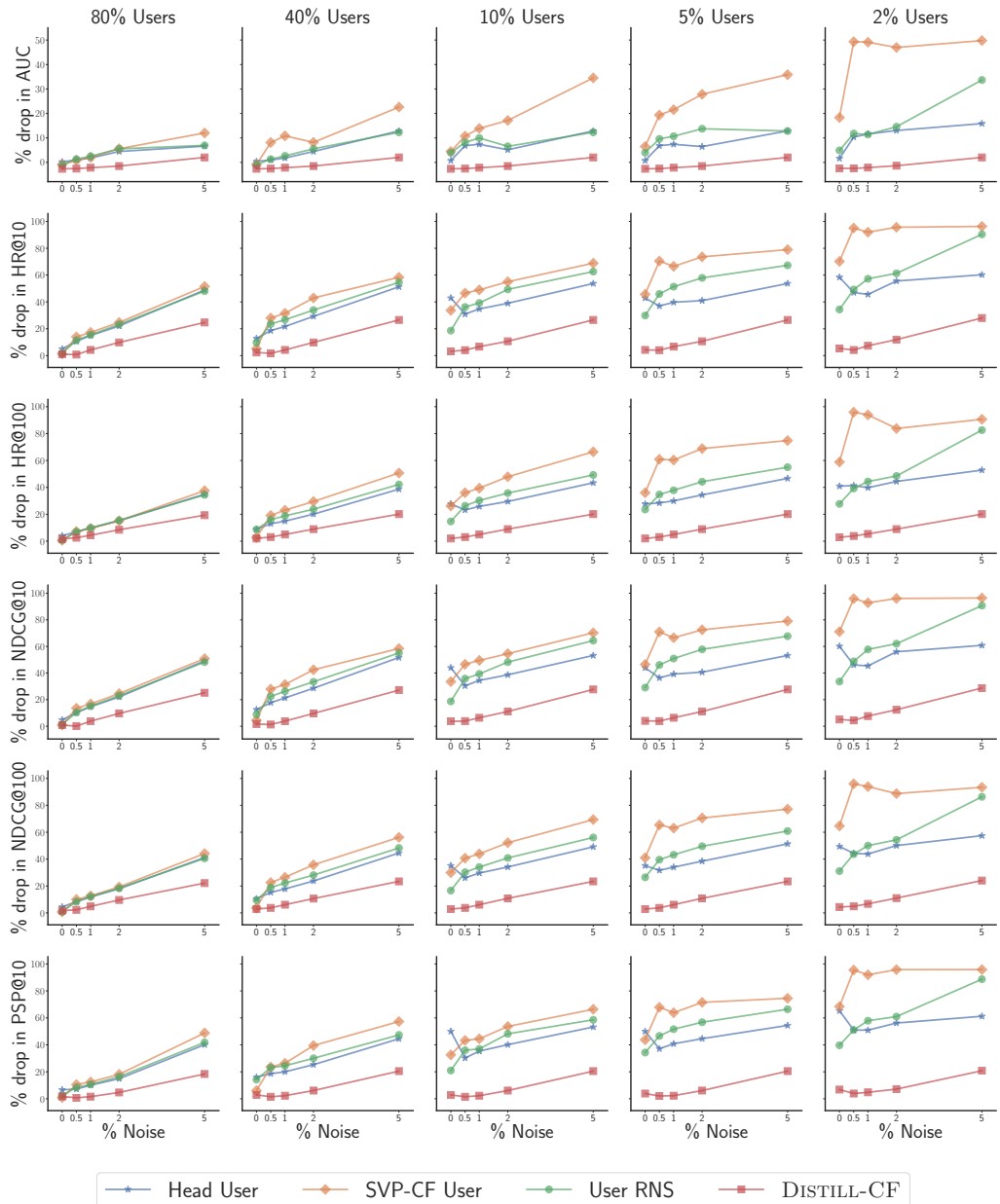

Figure 14: Performance of the EASE model trained on data sampled by different sampling strategies when there's varying levels of noise in the data. Each column represents a user sampling budget, and each row represents the % drop w.r.t a single evaluation metric. All results are on the MovieLens-1M dataset.

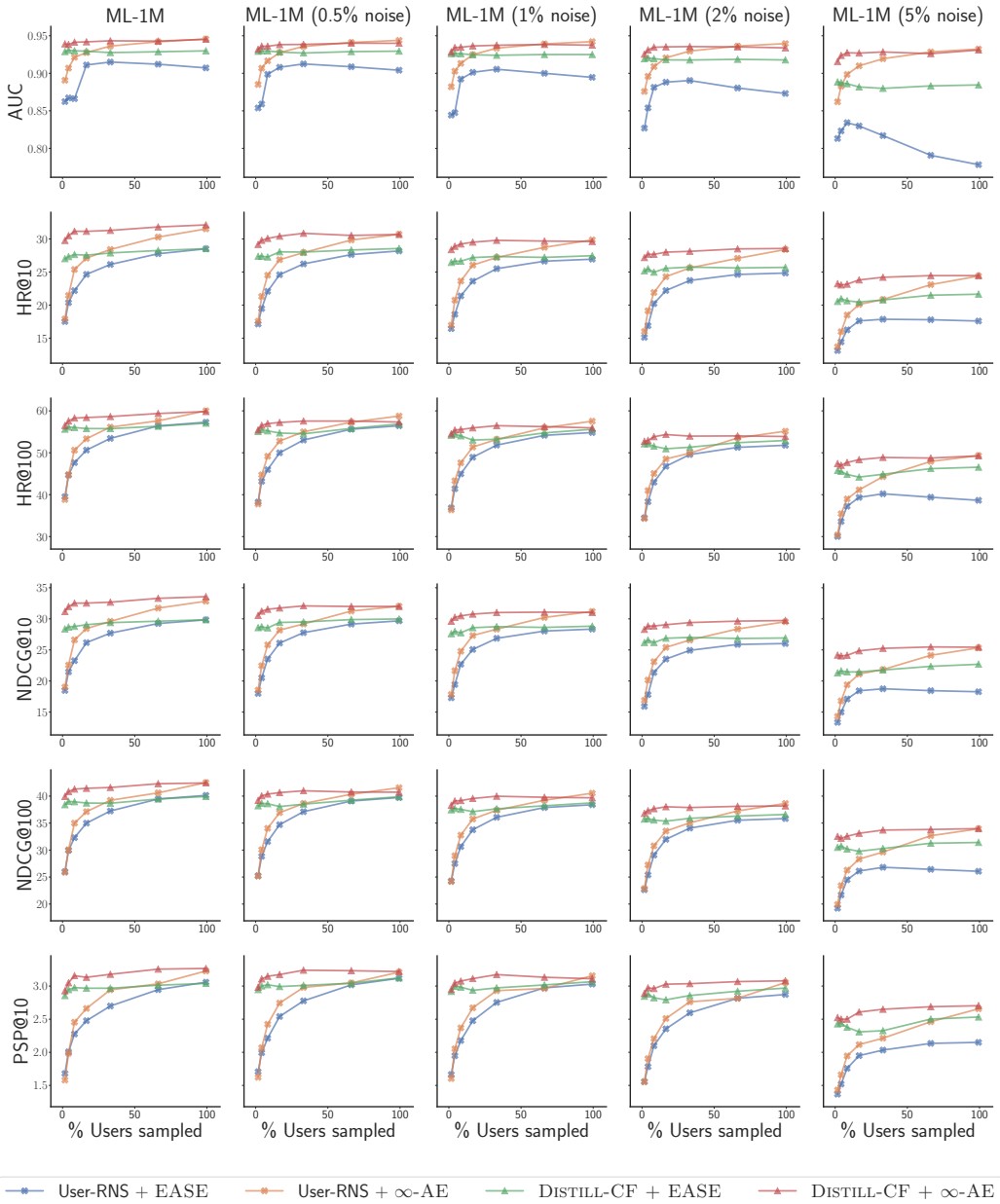

Figure 15: Performance of ∞-AE on data sampled by DISTILL-CF and User-RNS when there's noise in the data. Results for EASE have been added for reference. Each column represents a specific level of noise in the original data, and each row represents an evaluation metric. All results are on the MovieLens-1M dataset.