# OpenReview forum: "Infinite Recommendation Networks: A Data-Centric Approach"
_NeurIPS.cc/2022/Conference — NeurIPS 2022 Accept_

### Official Review · Reviewer_ScS5 · 2022-07-11

**Rating:** 5
**Confidence:** 3
**Soundness:** 3 good
**Presentation:** 2 fair
**Contribution:** 2 fair

**Summary:**

This paper leverages Neural Tangent Kernel and demonstrates the effectiveness of infinite-width autoencoders in recommendation systems. Furthermore, the authors also propose a data distillation framework to synthesize a small support dataset which aims to match the performance of algorithms trained on the full dataset via the bilevel optimization. Experiments are conducted on four public datasets to evaluate the performance of infinite-width autoencoders and the data distillation framework.

**Questions:**

Please refer to the detailed comments

**Strengths And Weaknesses:**

Strengths:
1. Provide an interesting view in enhancing recommendation systems by synthesizing a small but informative dataset, which could inspire more related work.
2. The proposed approach shows promising performance in improving recommendation performance, and meanwhile it is more robust to noise compared to SOTA.

Weaknesses:
1. To enhance the readability of the paper, it is highly recommended to provide more background information and preliminary content on NTK and KRR to help the audiences better understand the proposed approach as these are pretty new topics in recommendation systems.
2. If Distill-CF is claimed to be an general data distillation framework effective for CF datasets, it would be necessary to show how it performs while with other collaborative filtering models besides $\infty$-AE, e.g.,  how the synthesized data summaries work with other autoencoder-based recommendation systems?
3. The numbers of users are on different scales (e.g., 3k vs 476k). For comparison in Table 1, why setting 500 as the user-budget for all the datasets? Is it setting the user-budget based on the percentage of user number can help the audiences better understand the results?

---

> ### Author Response · Authors · 2022-08-02
> **Clarifications**
>
> Thank you so much for going through the paper and providing valuable feedback! Below are some clarifications which we hope will help in a better understanding and hopefully positively lean you towards recommending acceptance.
>
> **“To enhance the readability of the paper, it is highly recommended to provide more background information and preliminary content on NTK and KRR to help the audiences better understand the proposed approach as these are pretty new topics in recommendation systems.”**
>
> Thank you for the suggestion, we will include a primer on NTKs and KRR in the appendix in the final version. We understand that a reader in recommender systems might not be familiar with these concepts and a gentle introduction will be very relevant and useful.
>
> **“If Distill-CF is claimed to be an general data distillation framework effective for CF datasets, it would be necessary to show how it performs while with other collaborative filtering models besides -AE, e.g., how the synthesized data summaries work with other autoencoder-based recommendation systems?”**
>
> You’re right and there are two ways to go about this question, we’ll cover both of them, but let’s start with a clarification. The data summaries generated by Distill-CF have a different number of fake users (generally orders of magnitude smaller) which are used for model training. Remember that, models like NeuMF, LightGCN, etc. all are user-specific i.e. they need to see the same user in the training set for them to be able to recommend. On the other hand, autoencoder-based models (MVAE, EASE, ∞-AE) are user-free i.e. don’t need this assumption of seeing the same user in the train set, while also being more accurate. Hence, for our task, to evaluate the original test-set while training on the (small) data summary generated by Distill-CF, we can only train/test with user-free models (which we think you suggested in your question as well). Coming back to your question, there are two ways to go about it:
>
> (1) Using different models in Distill-CF’s inner loop: As we mentioned in lines 187-190, for Distill-CF to be computationally feasible while not sacrificing data quality, we need the inner loop to be as efficient and accurate as possible. ∞-AE fits these desiderata perfectly as its solution can be computed in closed-form with a single hyper-parameter, and is also better performing than other competitors. On the other hand, neural models (e.g. MVAE) typically tend to have a large number of hyper-parameters to tune at each iteration of the inner loop; while also being less accurate than ∞-AE (as demonstrated by our experiments). The argument holds valid for EASE as well, since ∞-AE is more accurate than EASE.
>
> (2) Evaluating different models on data optimized by using ∞-AE in Distill-CF’s inner loop: You might have missed it, but we did compare the robustness and generality of data synthesized using Distill-CF (optimized with ∞-AE) by training and evaluating the EASE model (Figure 2). We observed the same sampling: performance trends as for ∞-AE. In the hope to make the results even more prominent, during the rebuttal phase, we also trained and evaluated the MVAE model on data synthesized using Distill-CF (optimized with ∞-AE). From Figure 10 in the updated paper, we note the same trends as EASE and ∞-AE, validating our claim of data robustness and data generality.
>
> **“The numbers of users are on different scales (e.g., 3k vs 476k). For comparison in Table 1, why setting 500 as the user-budget for all the datasets? Is it setting the user-budget based on the percentage of user number can help the audiences better understand the results?”**
>
> Thank you for bringing up a valid point regarding the presentation of our results. We would firstly like to clarify that the percentage sampling-based results are already presented in Figures 1, 2, 10, and 13. Coming back to the main results in Table 1, we picked 500 fake users as a heuristic upper bound on the user budget needed for getting reasonable performance on most datasets. This practice has originated from the original, computer vision based dataset distillation papers, where they use the absolute number of images-per-class sampled [1, 2, 3].
>
> [1] Dataset Condensation with Gradient Matching. ICLR ‘21.
>
> [2] Dataset Condensation with Differentiable Siamese Augmentation. ICML ‘21.
>
> [3] Dataset Distillation with Infinitely Wide Convolutional Networks. NeurIPS ‘21.

---

### Official Review · Reviewer_s1k3 · 2022-07-13

**Rating:** 4
**Confidence:** 4
**Soundness:** 2 fair
**Presentation:** 2 fair
**Contribution:** 2 fair

**Summary:**

This paper proposed an autoencoder model where Neural Tangent Kernel is applied to estimate an infinitely-wide neural model named $\infty$-AE. Theoretically, infinitely-wide networks could bring highly expressive performance but still struggle with the large size of data in the recommendation. The author then proposed a learnable method for dataset summarization to tackle the problem. A lot of experiments are conducted to prove that the sampling method is effective and robust.

**Questions:**

In addition to the weakness mentioned above, the following questions are expected to answer.

- It's a little confusing that "the synthesized data was never be optimized (in the inner-loop of DISTILL-CF)" in line 279. Does it mean that $X^s$ in equation 3 is never optimized (but it notes the "inner-loop", which may indicate that the EASE is never optimized according to the definition of inner loop in line 185)? Maybe it should be "outer-loop" here. Perhaps you mean that the data distilled from well-trained Distill-CF can train any different models.
- In figure 3, is there any reason why the SVP-CF method suffers the most severe drop?
- In figure 4, the vertical axis is set as HR@10 instead of 'drop in HR@10' used in figure 3. It is not so convincing that $\infty$-AE is more robust than EASE, maybe the drop in $\infty$-AE is higher than EASE. And only RNS and Distill-CF methods are used, will $\infty$-AE get similar results when using "Head User" or "SVP-CF" method?
- In continual learning, the data from previous periods should be unreachable again, so the "joint" method can not be regarded as continual learning.
- It is better to explain how to combine MVAE with Distill-CF in continual learning experiments. Is that similar to the setting in figure 2, where the synthesized data is never optimized?
- In figure 5, the ADER method gets even worse performance than individual methods, which is not so convincing.
- In figure 6, the results on dataset ml-1m are totally different from another two datasets, where the deeper network gets much better performances.
- In Appendix A Algorithm 2, the input $\hat{X}_u$ should be noted in Input part: which may be "history vector of user to be predicted".


**Limitations:**

Not discussed

**Strengths And Weaknesses:**

- Strengths
    - The paper applied NTK method to gain an infinitely-wide networks which is more expressive.
    - The proposed data summarization method is actually a model-agnostic method, where the model is used to predict the user-item score in order to optimize the prior-matrix $X^S$.
   - The combination of $\infty$-AE and Distill-CF provide a simple but effective method to get a tiny model without loss of performance.


- Weakness
   - The Distill-CF method could be combined with different neural models which can be optimized alternately, such as NeuMF, EASE, LightGCN et al. However, the author didn't try it to verify the general effectiveness of Distill-CF.
   - Many baselines may not fined-tuned. For example, the baseline EASE is not fine-tuned, since the regularization coefficient is of vital importance.
   - The reason why the proposed method works well when only a small number of summarized data is used is not explained from a theoretical perspective.
   - Why the use of summarized data can even perform better than the use of all data in many cases?
  - The formulation of Gumbel softmax should be better described, otherwise, it is difficult to follow.
  - Several hyper-parameters should be fine-tuned in the proposed methods.
  - How to do inference? Only use the summarized data?
  - It is challenging to optimize the summary of the dataset, due to the use of the hard tanh function, and Gumbel sampling trick as well as the inverse of a parameter-dependent square matrix.

---

> ### Author Response · Authors · 2022-08-02
> **Clarifications (part 2)**
>
> **“It's a little confusing that "the synthesized data was never be optimized (in the inner-loop of DISTILL-CF)" in line 279. Does it mean..."**
>
> Apologies for the confusion, in our statement we are pointing out that to better understand the universality of data summaries, we train the EASE model on data synthesized by Distill-CF + ∞-AE. Our statement of “the synthesized data was never for it” was intended to signify that since the EASE model isn’t used in the inner-loop of Distill-CF, the data is not optimized for training EASE on the distilled data. Yes, you’re right: “data distilled from well-trained Distill-CF can train any different models” is exactly what we’re trying to show with this experiment.
>
> **“In figure 3, is there any reason why the SVP-CF method suffers the most severe drop?”**
>
> We hypothesize the reasons to be as follows: (1) SVP-CF is a coreset mining strategy intended to retain the “relative ordering of recommendation algorithms trained on the sampled data”. On the other hand, this paper intends to sample data in such a way that algorithms trained on the sampled data can match the performance compared to training on the full dataset. We note these two objectives to be relatively orthogonal and hence the mismatch. (2) Additionally, since SVP-CF relies on training a proxy model for down-sampling data, in the case of noisy data, a lot of noisy data points WILL be kept because they will tend to incur the most amount of loss while training the proxy model, and hence will be tagged as highly important.
>
> **“In figure 4, the vertical axis is set as HR@10 instead of 'drop in HR@10' used in figure 3. It is not so convincing that -AE is more robust than EASE, maybe the drop in -AE is higher than EASE. And only RNS and Distill-CF methods are used, will -AE get similar results when using "Head User" or "SVP-CF" method?”**
>
> We believe having an absolute comparison between ∞-AE and EASE is exactly the intent of this experiment. Instead of comparing the respective relative drops of ∞-AE and EASE, a practitioner will be more interested in looking at what’s the best model someone should use when the data is noisy. Reframing the experiment’s intent – at high noise ratios, is ∞-AE still the best model to use? As for using ∞-AE with other sampling strategies, we only include the most popular sampling strategy (random sampling) as well as our proposition (Distill-CF) for the sake of brevity and clarity.
>
> **“In continual learning, the data from previous periods should be unreachable again, so the "joint" method can not be regarded as continual learning.”**
>
> You’re correct, in the typical continual learning setup, data from previous periods is not reachable. However, in the ADER paper [1], which we use as the main reference and baseline, the authors compared ADER with this “joint” baseline. To fend off any criticism and lack of comparison with previous papers, we compared with this baseline as well.
>
> [1] Ader: Adaptively distilled exemplar replay towards continual learning for session-based recommendation. RecSys ‘20.
>
> **“It is better to explain how to combine MVAE with Distill-CF in continual learning experiments. Is that similar to the setting in figure 2, where the synthesized data is never optimized?”**
>
> Apologies for not being clear, and thanks for pointing it out – we have added a clarification in line 328 for this experiment that the data summary synthesized by Distill-CF is indeed optimized with ∞-AE in the inner-loop and not for MVAE (as you rightly mentioned the similarity with the setting in Figure 2).
>
> **“In figure 5, the ADER method gets even worse performance than individual methods, which is not so convincing.”**
>
> There are two explanations for your skepticism behind this observation: (1) the dataset we use (ML-1M) most certainly has different characteristics than the ones used in the original paper, and (2) we found a bug in the continual data creation in the author’s original implementation. Both these reasons result in ADER not performing well for continual learning.
>
> **“In figure 6, the results on dataset ml-1m are totally different from another two datasets, where the deeper network gets much better performances.”**
>
> As is expected in any machine learning application, the network architecture highly depends on the dataset and its characteristics. Similarly, the ML-1M is a dense dataset, where higher-order MLPs might be useful for performance. Nonetheless, having only a single layer of ∞-AE, even for the ML-1M dataset results in a maximum 1-2% drop in performance according to any metric.
>
> **“In Appendix A Algorithm 2, the input should be noted in Input part: which may be "history vector of user to be predicted".”**
>
> Yes, you’re right, thanks for pointing it out! We fixed it in the latest rebuttal version.

---

> ### Author Response · Authors · 2022-08-02
> **Clarifications (part 1)**
>
> Thank you so much for going through the paper and providing valuable feedback! Below are some clarifications which we hope will help in a better understanding and hopefully positively lean you towards recommending acceptance.
>
> **“The Distill-CF method could be combined with different neural models which can be optimized alternately, such as NeuMF, EASE, LightGCN et al. However, the author didn't try it to verify the general effectiveness of Distill-CF.”**
>
> We request you to kindly check the second clarification in our response to Reviewer ScS5 for a detailed clarification. We added Figure 10 in the updated paper for your question.
>
> **“Many baselines may not fined-tuned. For example, the baseline EASE is not fine-tuned, since the regularization coefficient is of vital importance.”**
>
> We strongly agree that the regularization coefficient in EASE is indeed highly important to tune. Knowing this observation, for all models (including EASE), we already conducted a sizable baseline hyper-parameter search for all experiments. Please check Table 3, Page 17 where we have already listed all the hyper-parameter configurations tried for both our models as well as the baselines.
>
> **“The reason why the proposed method works well when only a small number of summarized data is used is not explained from a theoretical perspective”**
>
> We agree that investigating the theoretical properties of Distill-CF (e.g. bias upper bounds of trained models, robustness, the convergence of trained models, etc.) is an interesting topic that we would like to investigate in future work. However, we would like to point out that “showing” this property, that less but high-quality data can be more useful even for long-tailed, sparse, and semi-structured data like Recommender Systems, as well as showing how to optimize for such data measures is an important contribution in itself.
>
> **“Why the use of summarized data can even perform better than the use of all data in many cases?”**
>
> We agree that this is a rather counterintuitive observation, for which we have a simple hypothesis – data quality is more important than data quantity for better model training. For example, in a case where we have large amounts of user-item interaction data available, if the data is noisy, it becomes harder to learn meaningful user behavior patterns rather than having lower amounts of high-quality, denoised data (which Distill-CF notably optimizes for). Note that, we observe better performance with lesser data only in some cases e.g. Table 1 (Amazon Magazine), Figure 4 (5% noise, for EASE), whilst in other experiments, using all data is indeed the best choice.
>
> **“The formulation of Gumbel softmax should be better described, otherwise, it is difficult to follow.”**
>
> Thank you for the suggestion! We clarified more in-depth about the Gumbel sampling procedure we follow, formally in Appendix B.4, Page 17.
>
> **“Several hyper-parameters should be fine-tuned in the proposed methods.”**
>
> ∞-AE has only a single hyper-parameter thanks to its closed-form optimization, which is a big bonus, especially in comparison to other state-of-the-art recommender systems which typically need a lot of hyper-parameter tuning for optimal performance. On the other hand, Distill-CF has only five hyper-parameters, which is similar (or in cases even less) in number to recent recommendation algorithms being published, albeit we are now optimizing for data sub-samples and not just estimating the parameters of learning algorithms on a fixed dataset.
>
> **“How to do inference? Only use the summarized data?”**
>
> Yes, we only need the summarized data for inference if we’re using ∞-AE with Distill-CF. To be precise, we only need the dual variables $\alpha$ estimated on the summarized data (line 147); and during inference, we can use the precomputed $\alpha$ along with the NTK of the evaluation user (Algorithm 2, Appendix A) to get score estimates.
>
> **“It is challenging to optimize the summary of the dataset, due to the use of the hard tanh function, and Gumbel sampling trick as well as the inverse of a parameter-dependent square matrix.”**
>
> While you raise a valid concern, we would like to clarify that Distill-CF is not hard to optimize. Hard-tanh can be seen as a variation of clamping, which is commonly used in a lot of deep learning implementations (e.g. gradient clipping). The same holds for Gumbel sampling as well, which finds its usage in a lot of reinforcement learning and text-based applications. As for the inverse operation, we would like to point out that the gramian matrix that needs to be inverted (line 146) is positive definite, guaranteeing the existence of the inverse. In addition to this, even in our experiments, we didn’t run into any stability issues. This behavior is demonstrated in our experiments, as we need only a single seed value for stable and optimal performance (line 253).

---

> > ### Comment · Reviewer_s1k3 · 2022-08-08
> > **More comments.**
> >
> > Thanks for the detailed response. I have more suggestions.
> >
> > **Hyperparameter tunning of baselines**
> >
> > Already reading the settings of hyperparameters, the authors are suggested to further fine-tune them, since they are so sensitive to the performance of some baselines.
> >
> > **Difficult training and theoretical understanding**
> >
> > I mean the gradient estimation of parameters is biased (due to hard tanh, BTW, how to estimate the gradient of hard tanh) and of large variance (Gumbel sampling) and maybe unstable (matrix inverse). It is absolutely we can reduce the loss by gradient descent, but how much can it reduce? what's the gap between the converged points and optimum? If we have the optimal sketch, how the proposed model can perform? There is no answer.
> >
> > Generally, this paper is promising. The paper should be further improved by addressing some important concerns.

---

> > > ### Author Response · Authors · 2022-08-09
> > > **Clarifications to your latest comment**
> > >
> > > Thanks for going through the rebuttal and comments! Here is our response to your latest comments:
> > >
> > > **Hyperparameter tunning of baselines**
> > >
> > > We want to point out that seeing Table 3, we search for up to 200 hyper-parameters for each of our baselines, which we believe is not at all common, and is a step we already took to conduct a fair evaluation. As for EASE, we would like to point out that we search for the same number of the semantically similar hyper-parameter $\lambda$ for our proposition $\infty$-AE as well -- making the comparison fair again.
> > >
> > > **Difficult training and theoretical understanding**
> > >
> > > As for difficult training, first, regarding the gradient of hard-tanh: it can be estimated using a piecewise function as described in [Page11, 1]. Regarding the unstable inverse, we would like to point out that adding $\lambda \cdot I$ before inverting makes the matrix provably PD, and practically, we don't observe any instability as proven by our methods' results being only over a single seed. We understand the obstacles you mentioned about Gumbel sampling, but we would like to point out that the bias-variance tradeoff can be controlled by the temperature term [2], and we also note that we take multiple Gumbel samples per user (row) followed by a union procedure (hard tanh), reducing the instability compared to the typical usage of Gumbel softmax -- taking a single Gumbel sample per distribution.
> > >
> > > About the difference to the optimal point, I'm assuming you are talking about Distill-CF, since $\infty$-AE already has a closed-form optimization. As for Distill-CF, we don't have an optimal solution as we're optimizing for the binary logistic loss in the outer loop. Hence, we don't have a way to comment on the gap between the converged points and the optimum. On a further note, your question boils down to the optimality of typical SGD optimization, which we believe is far from the scope of this paper.
> > >
> > > [1] https://cs224d.stanford.edu/lecture_notes/LectureNotes3.pdf
> > >
> > > [2] Categorical Reparameterization with Gumbel-Softmax. ICLR '17

---

### Official Review · Reviewer_3xoT · 2022-07-15

**Rating:** 7
**Confidence:** 4
**Soundness:** 4 excellent
**Presentation:** 3 good
**Contribution:** 3 good

**Summary:**

This paper develops an infinite-width autoencoder for recommendation. The proposed recommendation model contains a single hyper-parameter and only needs a closed-form solution. Leveraging -AE's simplicity, this paper develops Distill-CF for large data. According to the experiments on several datasets, the proposed model shows better performance.

**Questions:**

N/A.

**Limitations:**

N/A.

**Strengths And Weaknesses:**

strengths.
The proposed model is novel and interesting.
Compared with several baselines, the proposed model performs better.

---

> ### Author Response · Authors · 2022-08-02
> **Thank you for recommending acceptance!**
>
> Thank you so much for going through the paper and recommending acceptance – we are grateful to you for understanding the potential impact and application of ∞-AE and Distill-CF into various applications of recommender systems.

---

### Official Review · Reviewer_NjNV · 2022-08-27

**Rating:** 6
**Confidence:** 3
**Soundness:** 3 good
**Presentation:** 2 fair
**Contribution:** 3 good

**Summary:**

The paper studies a novel recommendation framework which combines two complementary ideas. The infinite-width autoencoder, $\infty$-AE, models the recommendation data, while DISTILL-CF creates a small set of data "summaries" (synthetic examples constructed from the real ones) used further for model training. The presented empirical results look very promising, and the introduced ideas are worth attention.

**Questions:**

As this is an additional review, I do not have any direct questions to the authors. Nevertheless, I would like to give some recommendations:
- Try to improve writing to make the paper less dense,
- Properly introduce, define, and cite important concepts,
- Properly justify your choices and claims,
- Extend the discussion on performance on tail items,
- Try to answer the question, why the small number of synthetic instances suffices to obtain good results,
- Check the code for possible bugs in the experiments.

**Limitations:**

- Limited experimentation with tail items

Although Table 1 and Figure 1 report the results for a metric somehow suited for tail items (PSP might be used for this purpose, but in general this is not the best choice because of its flaws; see, for example, "On Missing Labels, Long-tails and Propensities in Extreme Multi-label Classification", KDD 2022), the other experiments report HR@k which rather focuses on head items. I could overlook something, but an extended and more precise discussion should be given in the final version of the paper

- Self-criticism

I did not find an honest discussion about the flaws of the method. I would appreciate a more critical thinking from the authors


**Strengths And Weaknesses:**

Strengths:
- A though-provoking and novel (according to my knowledge) framework for recommendation combining relatively new, state-of-the-art concepts such as infinite-width networks, data summaries, and Gumbel sampling trick,
- Promising empirical results in terms of predictive and computational performance,
- Relatively wide experimental studies.

Weaknesses
- The paper is quite dense, therefore does not read very well; one can be easily lost,
- The paper is not self-contained; many concept are not properly introduced or cited (e.g., dual activation is neither defined nor cited; if not wrong, this paper should be used as reference: "Toward Deeper Understanding of Neural Networks: The Power of Initialization and a Dual View on Expressivity"),
- An extended discussion around the question "Is more data what you need for recommendation?" should be given in the paper. The results given in Paragraph "How robust are DISTILL-CF and $\infty$-AE to noise?" might be an effect of a bug in the experiments. Another reason could be the focus on head items as HR@k rather ignores tail items. Figure 1 reports promising results for PSP@k, but this might not be the best metric to use for tail items (see below).

Minor comments:
- "infinitely-wide bottleneck layers" sounds contradictory: This design choice needs more discussion to justify the approach. Is the regularization "responsible" for the bottleneck?
- It is written: "We can subsequently perform inference for any novel user as follows (...)". The novel users will have no history, so their representation will be empty. This part needs additional comments.
- I am not familiar with the recent sota approaches for recommendation, so it might be that the experimental results are less impressive than given,
- Function $f$ in (1) should be properly defined to make it clear that its value is a vector,
- $D^{val}$ seems to be not defined,
- Use "\log" in math formulas for logistic function.

---

### Meta-Review · Area_Chair_KXeu · 2022-08-29

**Recommendation:** Accept
**Confidence:** Less certain

**Metareview:**


This paper proposes an infinite-width autoencoder for recommendation, which trained using the NTG framework by the kernelized ridge regression algorithm. The approach struggle with the large size of the data. To this end, the authors propose a method for data set summarization, called Distill-CF, for synthesizing tiny high-fidelity data summaries.


The paper received a mixed evaluation from the reviewers.

The strengths of the paper mentioned by the reviewers were:
- A simple model with only one hyper-parameter and a closed-form solution
- A though-provoking and novel framework for recommendation
- Good performance in the experiments, relatively wide experimental study

On the other hand, the identified weaknesses were:
- The author did not verify the general effectiveness of Distill-CF beyond coupling with infinite AE, so it is not clear where is the actual gain
- Technical issues with the experiments
- Some issues with the readability

**Award:**

No

---

### Decision · Program_Chairs · 2022-09-14

Accept